# How Good is Post-Hoc Watermarking With Language Model Rephrasing?

**Pierre Fernandez** [* 1]   **Tom Sander** [* 1]   **Hady Elsahar** [1]   **Hongyan Chang** [1]   **Tomáš Souček** [1]   **Valeriu Lacatusu** [1]
**Tuan Tran** [1]   **Sylvestre-Alvise Rebuffi** [1]   **Alexandre Mourachko** [1]

## Abstract

*Generation-time* text watermarking embeds statistical signals into text for traceability of AI-generated content. We explore *post-hoc* watermarking, where an LLM rewrites existing text while applying watermarks to protect copyrighted documents or detect their use in training and RAG via watermark radioactivity. Unlike generation-time approaches constrained by serving infrastructure, this setting offers additional flexibility for both generation and detection. We thus investigate how compute allocation (larger models, beam search, multi-candidate generation, entropy filtering) affects the quality-detectability trade-off. We find that the simple Gumbel-max scheme surprisingly outperforms recent alternatives under nucleus sampling, achieving strong detectability and semantic fidelity on open-ended text like books. Most methods benefit significantly from beam search, and counterintuitively, smaller models outperform larger ones. However, this approach struggles with verifiable text such as code. Code is available at https://github.com/facebookresearch/textseal.

## 1 Introduction

Post-hoc text watermarking inserts an algorithmically-detectable signal into an existing text while remaining imperceptible to readers. It serves several goals, such as copyright protection or traitor tracing. Early approaches relied on hand-crafted modifications: synonym substitutions (Topkara et al., 2006c; Shirali-Shahreza & Shirali-Shahreza, 2008), grammatical transformations (Topkara et al., 2006a;b) or morphosyntactic alterations (Meral et al., 2009). Recent methods based on deep neural networks follow a post-hoc paradigm where the input is the original text and the output

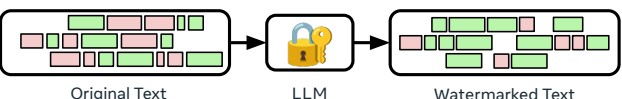

*Figure 1.* **Post-hoc text watermarking through watermarked LLM rephrasing.** We do empirical evaluations and analyze detection power, semantic fidelity, and correctness according to different design choices such as watermark scheme and available compute (through the paraphrasing model and decoding strategy).

is the watermarked text, using one watermark embedder and one watermark extractor (Abdelnabi & Fritz, 2021), similarly to state-of-the-art post-hoc watermarking approaches for images, audio, and video. However, these methods are not effective, suffering from low capacity or unreliable detection, because they are highly constrained or easily broken by reversing the edits or synonyms.

With the emergence of large language models (LLMs), their popularization through ChatGPT (OpenAI, 2022), and growing concerns about potential risks (Crothers et al., 2022; Weidinger et al., 2022), generation-time text watermarking algorithms have been introduced to help detect AI-generated text (Aaronson & Kirchner, 2023; Kirchenbauer et al., 2023a). These algorithms have been deployed at scale, for instance, in Google's Gemini with SynthID (Dathathri et al., 2024). Most LLM watermarks alter the next token selection, for example by promoting a specific set of tokens depending on previous tokens and a secret key. Detection of the watermark in a text is then performed through a theoretically grounded statistical test that provides rigorous guarantees over the false positive rates. In addition, these methods achieve high watermark power while adding minimal latency by leveraging the entropy of the LLM to embed the signal. However, in contrast with previous approaches, these methods require control at generation time and therefore cannot be applied as such to existing text.

A natural idea to extend these methods to post-hoc watermarking is to employ a paraphrasing LLM to re-generate the text, allowing the watermark to be injected at inference time without further model training. This approach has been explored in the data protection literature (Jovanović et al., 2025; Sander et al., 2025; Rastogi et al., 2025; Zhang et al., 2025; Lau et al., 2024). For example, "watermark radioactivity" exploits the fact that watermark signals leave

---

[*]Equal contribution [1]FAIR, Meta Superintelligence Labs. Correspondence to: Tom Sander <tomsander@meta.com>, Pierre Fernandez <pfz@meta.com>.

*Proceedings of the 43rd International Conference on Machine Learning*, Seoul, South Korea. PMLR 306, 2026. Copyright 2026 by the author(s).

detectable traces when watermarked text is used by another model, enabling active training and context membership inference (Sander et al., 2025; Jovanović et al., 2025). However, to our knowledge, there is no thorough evaluation of how post-hoc watermarking through LLM paraphrasing performs across different data domains, such as prose versus verifiable text like code. Moreover, unlike generation-time watermarking, where watermark embedding should not delay the generation, post-hoc watermarking allows trading additional computation for a better quality-detectability trade-off. This opens up several axes to explore: model size, watermarking method, decoding strategy (and even generating multiple candidates), as well as detection strategy.

To address this gap, we provide a comprehensive evaluation of post-hoc watermarking. We find that current state-of-the-art LLM watermarking schemes applied in a post-hoc setting perform well on open-ended text, such as books and Wikipedia articles, achieving high detection power while preserving fidelity. However, these methods are less effective on verifiable text like code, where the requirement to preserve correctness severely constrains paraphrasing freedom. Regarding design choices, we find that larger models better preserve semantics, while smaller models are necessary to effectively hide strong watermarks. Furthermore, the simplest Gumbel-max approach (Aaronson & Kirchner, 2023) dominates the Pareto frontier when classical random sampling is used. For other watermarking methods, we find that beam search decoding significantly improves the quality-detectability trade-off. Our main contributions are:

- *A comprehensive evaluation:* We conduct the first large-scale study of post-hoc watermarking through LLM rephrasing across diverse domains, demonstrating that while current methods are effective for open-ended text, they struggle with verifiable formats like code.

- *An analysis of design strategies:* We isolate the impact of model size, decoding, and watermarking schemes. For instance, we identify Gumbel-max with mid-size model as the robust Pareto-optimal choice (Figure 2, Figure 3).

## 2 Related work

### 2.1 Post-hoc Text Watermarking

Early text watermarking altered text characteristics like characters or spacing (Brassil et al., 1995). Other methods modify grammatical or syntactical structures via pre-established rules (Topkara et al., 2005), including synonym substitution (Topkara et al., 2006c) and word reordering through passivization or topicalization (Topkara et al., 2006b;a; Meral et al., 2009). Text steganography follows similar principles (Winstein, 1998; Chapman et al., 2001; Bolshakov, 2004; Shirali-Shahreza & Shirali-Shahreza, 2008; Chang & Clark, 2014; Xiang et al., 2017). These edit-based sys-

tems exhibit low robustness and payload, e.g., 1-2 bits per sentence (Wilson & Ker, 2016). Deep learning methods have since been used for this task, for instance, with masked language models for steganography (Ueoka et al., 2021), infilling models (Yoo et al., 2023a), neural lexical substitution (Qiang et al., 2023), or encoder-decoders (Abdelnabi & Fritz, 2021; Zhang et al., 2024; Xu et al., 2024).

### 2.2 Generation-Time Watermarking With LLMs

The first watermarks for machine-generated text date back to a method presumably used in Google Translate to filter their translations from future training data (Venugopal et al., 2011). For LLM-generated text, two concurrent approaches appeared shortly after the release of ChatGPT. (Kirchenbauer et al., 2023a) bias a subset of the vocabulary, while (Aaronson & Kirchner, 2023) alter the sampling via the Gumbel trick. Both use pseudorandom seeds generated from a secret key and preceding tokens, then enabling lightweight statistical detection without access to the model.

Subsequent work explores improved tests and multi-bit watermarking (Fernandez et al., 2023; Yoo et al., 2023b; 2024; Qu et al., 2024), position-dependent seeds (Christ et al., 2023; Kuditipudi et al., 2023), low-entropy optimizations (Lee et al., 2023; Christ et al., 2023; Huang et al., 2023), and semantic watermarks for better robustness (Liu et al., 2023; Liu & Bu, 2024; Fu et al., 2024; Hou et al., 2023; 2024). DiPMark (Wu et al., 2023) provides distortion-free Green-Red watermarks, and MorphMark (Wang et al., 2025) adaptively adjusts watermark strength based on the green token probability mass. WaterMax (Giboulot & Furon, 2024) generates several chunks of tokens from the original LLM distribution and selects outputs with high watermark scores, which ensures that the original LLM distribution is preserved. SynthID-Text (Dathathri et al., 2024) deploys tournament-based sampling in Google Gemini. Toolkits have also been introduced to benchmark these methods (Piet et al., 2023; Pan et al., 2024). We provide a comprehensive description of all the schemes evaluated in this work in subsection A.2.

### 2.3 Post-hoc LLM Watermarks for Data Protection

Recent works apply LLM watermarks to training or evaluation data via paraphrasing, similar to what we study in this work. Most exploit watermark radioactivity (Sander et al., 2024), i.e., the detectable traces left when watermarked text is used for training. Applications include detection of texts used in retrieval-augmented generation (RAG) (Jovanović et al., 2025), benchmark contamination detection (Sander et al., 2025), and training data copyright (Zhang et al., 2025). Only Waterfall (Lau et al., 2024) evaluates post-hoc watermarking through LLM paraphrasing, focusing on code (MBPP) and natural text (C4, arXiv) for provenance.

These works show the utility of post-hoc watermarking through LLM paraphrasing for data protection, but do not evaluate it as a general watermarking method, nor do they characterize failure modes across text types and settings.

## 3 Method

The idea is to rephrase the input text using an LLM while applying watermarking during generation ( Figure 1). Specifically, we: (1) Split text into chunks (sentences or paragraphs), (2) for each chunk, prompt an LLM to produce a paraphrase given specific instructions (e.g., preserve named entities) and previous context, (3) during decoding, sample the next token with watermarking: e.g. favor tokens in a "greenlist" according to a strength parameter or any method mentioned in subsection 2.2 and detailed in subsection A.2, and (4) aggregate watermark scores and compute a statistic.

### 3.1 LLM Watermarking

**Next token generation.** At each decoding step, the LLM computes logits $\ell \in \mathbb{R}^{|\mathcal{V}|}$ over the vocabulary $\mathcal{V}$ conditioned on the context. $\ell$ is then converted into a probability distribution $\mathbf{p} = \mathrm{softmax}(\ell/T)$ by using a temperature $T$.

**Watermarked sampling.** Watermarking modifies the decoding process rather than sampling directly from $\mathbf{p}$. This is done depending on a window $\mathbf{w}$ of the $k$ preceding tokens and a secret key $s$. A pseudorandom function $\mathrm{PRF}(\cdot)$ combines $\mathbf{w}$, $s$, and a candidate token $v$ to produce a value used for biasing the sampling of token $v$. For instance, in the green-red method (Kirchenbauer et al., 2023a), we compute $\mathrm{PRF}(\mathbf{w}, s, v)$ for all tokens $v \in \mathcal{V}$ to partition the vocabulary into a "greenlist" and a "redlist," then bias the logits by adding $\delta$ to green tokens. Other schemes alter the sampling mechanism directly (e.g., with the Gumbel-max trick (Aaronson & Kirchner, 2023)). This embeds a statistical signal into the generated text that is imperceptible to readers but detectable by the detector. We provide a comprehensive description of all schemes in subsection A.2.

**Statistical test.** Detection recomputes the pseudorandom function for each group of tokens ($\{k + 1\}$-grams) and aggregates the results into a test statistic $K$. Under the null hypothesis $\mathcal{H}_0$ "not watermarked with this specific scheme", $K$ should follow a known distribution, yielding a $p$-value for the probability of observing such or a more extreme score by chance. Under $\mathcal{H}_1$ "watermarked text with this specific scheme", the statistic should deviate significantly enough and lead to low $p$-values. We flag a text as watermarked if $p$-value $\leq \alpha$, which controls the false positive rate (FPR).

For instance, if $K$ (the test statistic) is the observed number of green tokens out of $N$ scored tokens, it should follow under $\mathcal{H}_0$ a Binomial distribution with parame-

ters $N$ and $\gamma$ (expected green ratio, typically 0.5). So $p$-value $= \mathbb{P}(X \geq K \mid X \sim \mathrm{Binomial}(N, \gamma))$. As an example, if we observe $K = 65$ green tokens out of $N = 100$ with $\gamma = 0.5$, the $p$-value is approximately $10^{-3}$; to flag this text as watermarked, we would need a FPR of $\alpha \geq 10^{-3}$.

### 3.2 Compute-Driven Flexibility

Operating in a post-hoc setting offers more flexibility than in-generation watermarks, which must comply with live-serving constraints such as latency, memory, and decoding speed (Dathathri et al., 2024). Here, we can use large or small models, run higher temperatures or beam search instead of simple sampling, and even employ multi-candidate selection like WaterMax (Giboulot & Furon, 2024), as a function of available compute and quality targets. We detail the specific higher-compute methods we test below.

**Beam search for watermarking.** We use a beam search to improve the quality-detectability trade-off. Specifically, we maintain $B$ sequences and expand each beam with $V$ candidates sampled from the watermarked probability distribution $\mathbf{p}_{\mathrm{wm}}(\cdot \mid x_{<t})$, which is derived by applying the watermarking scheme (e.g., Green/Red or SynthID detailed in subsection A.2) to the base model's logits. To select the top-$B$ beams at each step, we score candidates using log-probability under a reference distribution. We explore two scoring variants: *unbiased scoring* uses the original model probabilities $\mathbf{p}_{\mathrm{orig}}(\cdot \mid x_{<t})$ to favor sequences with lower perplexity relative to the base model, prioritizing text quality; *biased scoring* uses the watermarked probabilities $\mathbf{p}_{\mathrm{wm}}(\cdot \mid x_{<t})$ to favor sequences that are most probable under the watermarked distribution, potentially yielding stronger watermark signals. We note that the latter was also explored for generation-time watermarking in (Kirchenbauer et al., 2023b). For candidate generation, we consider deterministic beam search (selecting top-$V$ tokens by $\mathbf{p}_{\mathrm{wm}}$) and stochastic beam search (sampling $V$ tokens from $\mathbf{p}_{\mathrm{wm}}$).

**Multi-candidate selection with WaterMax (Giboulot & Furon, 2024).** Rephrasing is done chunk by chunk, with chunks of $L$ tokens. For each chunk, we generate $m$ candidates $\{\tilde{y}^{(1)}, \dots, \tilde{y}^{(m)}\}$ *without* applying any logit bias. We then select the candidate that naturally maximizes the watermark score: $y^* = \arg\max_y \mathrm{Score}_{\mathrm{wm}}(y)$. This method is "distortion-free" regarding the sampling distribution but is typically infeasible in standard API usage due to cost (generating $m$ times more tokens per chunk).

**Entropy-aware detection.** At detection time, we compute the entropy $H_t$ of a model's distribution at each token position $t$ in the watermarked text $H_t = -\sum_{v \in \mathcal{V}} p_t(v) \log p_t(v)$ where $p_t(v)$ is the model probabil-

ity for token $v$ given the prefix up to $t-1$ in the watermarked text. We then apply an entropy filter: a token is scored only if $H_t$ exceeds a chosen threshold $\tau$, similar to what is done in Lee et al. (2023) for code. This focuses detection on high-entropy positions, where watermarking is more effective. All entropy computations are performed only on the water-marked text, as the original text is not available at detection time. As a result, these entropy values differ from those at generation-time, when the model was conditioned.

## 4 Experiments

We evaluate post-hoc text watermarking, assessing detection power and fidelity. Experimental details are provided in subsection 4.1. We then compare watermarking methods on the quality-detectability Pareto (subsection 4.2), analyze the effect of model choice (subsection 4.3), beam search (subsection 4.4), and entropy-aware detection (subsection 4.5). Finally, we investigate watermarking code (subsection 4.6), cross-lingual robustness (subsection 4.7), and chunking for long documents (subsection 4.8).

### 4.1 Experimental Setup

**Datasets.** We evaluate on three diverse corpora to capture different text characteristics and use cases. (1) *Books:* passages in English from the Gutenberg dataset (Project Gutenberg, 2025), containing books. These represent longer-form, literary texts with complex sentence structures. (2) *Wikipedia:* The beginning of randomly sampled Wikipedia articles. These represent factual, encyclopedic texts with a formal writing style and different languages. (3) *Code:* Programming problems from the HumanEval (Chen et al., 2021) and MBPP (Austin et al., 2021) benchmarks, consisting of Python functions, with accompanying unit tests. For each problem, we concatenate the provided prompt (function signature and docstring) with the canonical solution. The dataset comprises 164 HumanEval and 974 MBPP samples, with lengths ranging from 20 to 500 tokens. Since watermarking may inadvertently rename function identifiers, we apply postprocessing to restore the original entry point name. Further details are provided in subsection B.2. An example task is shown in Figure 11 of the appendix.

**Paraphrasing models and prompts.** We experiment with several dense open-weight instruct LMs as paraphrasers, spanning different families and sizes to study the impact of model scale on paraphrase quality and watermark preservation. Specifically, we use the instruct version of Llama-3 (Dubey et al., 2024), Gemma-3 (DeepMind, 2025), Qwen-2.5 (Qwen Team, 2024), and SmolLM-2 (Allal et al., 2025). To ensure the model outputs the watermarked text directly and without any comments, we use a structured system prompt and prefill the assistant's response with a prefix (e.g., "Rephrased text:"). An example prompt used is: *"You*

*are a text rephrasing assistant. You must rephrase the text while preserving its meaning, style, and structure. You must output only the rephrased text, with no commentary."* For long documents, we employ a chunking strategy where each chunk is processed separately while optionally prepending previously rephrased chunks to preserve coherence.

**Watermarking schemes.** We evaluate multiple watermarking methods (detailed in subsection A.2), all with window size 2, varying their hyperparameters. (1) *Green-Red List (Kirchenbauer et al., 2023b):* bias $\delta \in \{1.0, 2.0, 4.0\}$. (2) *DiPMark (Wu et al., 2023):* reweighting parameter $\alpha \in \{0.2, 0.3, 0.4\}$. (3) *MorphMark (Wang et al., 2025):* $\kappa = 10$ fixed, and vary $p_0 \in \{0, 0.05, 0.1, 0.2\}$. (4) *SynthID-Text (Dathathri et al., 2024):* number of tournaments $k \in \{10, 20, 30\}$. We compute the $p$-values from weighted and non-weighted scores as described in subsection A.2. (5) *Gumbel-max (Aaronson & Kirchner, 2023):* fixed by temperature and top-$p$.

**Choice of private key.** We address the private key issue described in section 3 by testing 50 candidate keys. For each combination of tokenizer, watermarking method (and depth for SynthID), we evaluate the keys on non-watermarked texts to estimate the $p$-value distribution under $H_0$. We then select the key that minimizes the deviation from $U(0, 1)$, as measured by the Kolmogorov–Smirnov statistic. Next, we verify that the theoretical FPRs for these keys match the empirical FPRs on 1.5M Wikipedia documents (each contains up to 100k tokens, with an average of 715 tokens). This ensures that the subsequent results under $H_1$ reflect the performance of each scheme rather than artifacts of a particular key choice. Further details are provided in Table 5.

**Test validity.** Because the pseudorandom function depends on the preceding $k$ tokens, repeated n-grams generate identical hashes. This violates the statistical independence of the scores used when assuming the distribution of the test statistic under $\mathcal{H}_0$. Aggregating scores only over unique watermark windows within the text (Kirchenbauer et al., 2023a; Fernandez et al., 2023) is a good way to mitigate this issue and to ensure valid $p$-values. But deduplication alone may not guarantee valid statistical tests because natural language inherently favors certain n-grams. This can cause $p$-values to be artificially low or high even for unwatermarked text. To address this, we ensure that under $\mathcal{H}_0$, $p$-values are approximately uniform on $U(0, 1)$. In practice, we test many candidate secret keys and select one for which the empirical $p$-value distribution on unwatermarked text is close to uniform. We provide more details in subsection A.1.

**Entropy aware detection.** For all methods, we evaluate entropy-aware detection by computing per-token entropy

with the same model as used for rephrasing (see subsection 3.2). At detection, we only score tokens with entropy $\geq \tau$, with $\tau \in \{0, 0.2, 0.4, 0.6, \ldots, 1.8, 2.0\}$.

**Decoding strategies.** We explore different decoding approaches: (1) *Nucleus sampling (Holtzman et al., 2019):* we vary temperature $T \in \{0.7, 1.0, 1.2\}$ and top-$p \in \{0.9, 0.95, 0.99\}$. Higher temperature or top-$p$ should increase entropy at the cost of faithfulness. (2) *Beam search (Sutskever et al., 2014),* as detailed in section 3. We test beam widths $B \in \{3, 5, 10\}$ with the same number of candidates per beam. We experiment with selection criteria based on either the non-watermarked likelihood or the watermarked model's likelihood, and use stochastic (temperature 1 and top-$p$ 0.95) or deterministic selection within beams. (3) *WaterMax (Giboulot & Furon, 2024),* as detailed in section 3: for each draft chunk of $L \in \{4, 8, 16\}$ tokens, we generate $m \in \{4, 8\}$ non-watermarked candidate chunks, select the candidate with the highest watermark statistic, and repeat.

**Evaluation metrics.** For watermark detection, we compute the $p$-**value** of the corresponding detection test, which can be read as the probability that a watermark score at least as high would happen if the text were not watermarked. For rephrasing quality evaluation, we use three axes: (1) **SBERT** cosine similarity to measure meaning preservation between original and rephrased texts, using `all-MiniLM-L6-v2` for English and `paraphrase-multilingual-MiniLM-L12-v2` for multilingual evaluation, (2) average **cross entropy** of the watermarked text conditioned on the prefix "*This is a rephrased version of [ORIGINAL TEXT]:*" to capture if the output is fluent (using Mistral-7B-v0.3 (Jiang et al., 2023), an evaluator outside our paraphrasing model set), and (3) we sometimes report the character-level **length ratio** between rephrased and original text, which should remain close to 1.

In Sections 4.2, 4.3, 4.5, 4.4 and 4.8, each value corresponds to the median over 100 passages of approximately 800 characters (∼200 tokens) sampled from Charles Dickens' novels in the Gutenberg dataset (Project Gutenberg, 2025). We keep only rephrasings where the output length remains comparable to the original (length ratio $\in [0.75, 1.25]$), and report results only for settings where at least 70% of passages satisfy this constraint. As detailed in Appendix B, output length distributions are remarkably consistent across watermarking schemes and are primarily determined by the base model rather than the watermarking method. While output length is arguably a quality signal in itself, we invite the reader to examine these distributions in the appendix, allowing the main analysis to focus on semantic fidelity and detection power within valid generation bounds.

## 4.2 Quality-Detection Trade-Off

We first evaluate the fundamental trade-off between rephrasing quality and detection strength in Figure 2, using Llama-3.2-3B-Instruct with random sampling. Each point corresponds to a distinct parameter configuration as described in subsection 4.1. Optimal methods occupy the top-right region of the plot, combining high text quality with strong detectability. The Gumbel-max method appears to dominate this Pareto frontier, followed by MorphMark and SynthID. We observe the same ranking on Wikipedia and ELI5 passages, which suggests that the result is not specific to Dickens; see Table 7.

*We observe similar results after rephrasing attacks, and for perplexity instead of SBERT: see Figure 12 of Appendix B.*

We also present a qualitative example in Table 1, where we rephrase text using Llama-3.2-3B-Instruct with the Gumbel-max watermark with $T = 0.7$ and $p = 0.95$. The rephrasing maintains the original tone, structure, and factual content. Additional examples are provided in Table 14 of Appendix C.

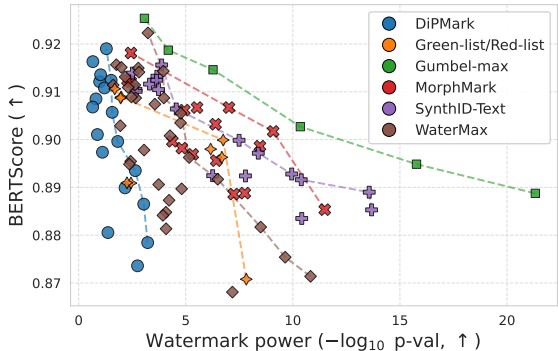

*Figure 2.* **Pareto fronts** showing the trade-off between quality and watermark strength using Llama-3.2-3B-Instruct. One point corresponds to one parameter configuration, and values are medians across 100 rephrased passages. Details are in Sections 4.1 and 4.2.

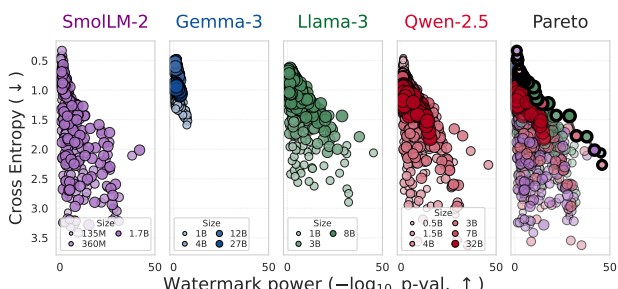

*Figure 3.* **Impact of model family and size.** Quality vs. watermark strength. Larger models improve quality for a given watermark strength, but small models are necessary to reach high strengths. All families are comparable, except Gemma-3 which is not suitable. Experimental details are given in Sections 4.1 and 4.3.

*Table 1.* **Qualitative comparison on literary text.** A sample from *A Christmas Carol* by Charles Dickens, rephrased using Llama-3.2-3B-Instruct with the Gumbel-max scheme, $T = 0.7$ and top-$p = 0.95$. The output maintains the original tone and structure while achieving high semantic similarity and good watermark detectability. The full text is given in Table 13.

| Original Text | Watermarked Rephrasing |
|---|---|
| [. . . ] "Why do you doubt your senses?" | [. . . ] "Why do you question the reliability of your senses?" |
| "Because," said Scrooge, "a little thing affects them. A slight disorder of the stomach makes them cheats. You may be an undigested bit of beef, a blot of mustard, a crumb of cheese, a fragment of an underdone potato. There's more of gravy than of grave about you, whatever you are!" [. . . ] | "Because," Scrooge said, "a minor aberration can significantly impact them. A slight irregularity in digestion can cause them to be misled. You may be an undigested morsel of meat, a splash of mustard, a fragment of a partially cooked potato. There's more of gristle than of flesh about you, regardless of what you are!" [. . . ] |

**Stats for full text shown in Table 13:** SBERT: 0.904 ● Perplexity: 2.61 ● $p$-val: $1.7 \times 10^{-5}$

## 4.3 Impact of Model Family and Size

We investigate whether larger models are better suited for post-hoc watermarking. Figure 3 shows the trade-off curves by model family (SmolLM, Gemma, Llama, Qwen) and size. Regarding performance, we observe that larger models preserve semantics better: across all families, increasing model size (darker points) shifts the clusters upward. For example, within the Llama family, Llama-3.1-8B (dark green) attains lower cross entropy than the 3B and 1B variants (lighter teal) at comparable watermark strengths. However, larger models struggle to reach low $p$-values compared to smaller models: they populate the left rather than the right of the plots. Gemma 3 models are essentially absent from the high-detectability region, as their outputs naturally exhibit very low entropy, even for the smallest sizes. Overall, when strong watermarking is required, only small models appear on the frontier, whereas larger models dominate in the lower-strength regime. We see in subsection 4.6 that the issue with large models persists even with higher temperature.

Note that these results are conditioned on the length filtering described in subsection 4.1; thus, the presence of a data point implies enough successful valid generation for the corresponding combination of model/scheme/parameters. For instance, smaller models with stronger watermarks fail to meet length constraints more frequently. Appendix B gives the generation stability of each configuration, which cannot be inferred directly from the figure.

## 4.4 Beam Search vs. Sampling vs. WaterMax

Figure 4 explores if beam search can find better watermarked sequences than random sampling. We watermark with Llama-3.2-3B-Instruct, and decode with or without beam search, for suitable watermarking method: Green-Red, SynthID, MorphMark, and DiPMark (since Gumbel-max is deterministic). We observe that beam search consistently improves the Pareto frontier, *especially with biased scoring* (see section 3): it shifts the results upward, improving quality at a fixed watermark strength. Notably, while Gumbel-max sampling appears to be the best option when used with random sampling in Figure 2, other methods can be substan-

tially improved through beam search. At matched quality, however, beam-enhanced alternatives still do not clearly surpass Gumbel-max; for example, Gumbel-max reaches $-\log_{10} p = 7.7$ at SBERT $0.862$, compared with $7.2$ at SBERT $0.872$ for MorphMark with biased beam search. WaterMax (Giboulot & Furon, 2024) results are shown in brown in Figure 2. Surprisingly, WaterMax achieves weak detectability across the hyperparameters we tested, making it difficult to achieve strong watermarking in our post-hoc setup.

## 4.5 Entropy-Aware Detection

Instead of scoring every token, we now filter out low-entropy tokens at detection time, as motivated in section 3 and detailed in subsection 4.1. In Figure 5, we fix the rephrasing model to Llama-3.2-3B and, for each method and watermarking configuration, test whether there exists an entropy threshold that improves detectability by more than $5\%$ on at least 50 of the 100 texts of Charles Dickens.

The left plot reports, for each watermarking method, the proportion of configurations for which such a threshold exists. We see that most methods have only 30–40 successful configurations. The right plot shows, for those successful configurations, the actual gain in detection performance, which never exceeds $20\%$.

We see that WaterMax is different: any entropy threshold degrades detectability. This is because WaterMax optimizes the watermark score at the *sentence* level rather than the

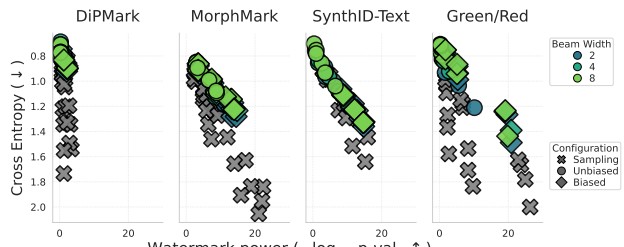

*Figure 4.* **Beam search improves the Pareto.** Cross entropy vs. watermark strength for suitable methods. Beam search, especially with biased scoring (see section 3), shifts the frontier upward, substantially improving rephrasing quality at a fixed watermark strength. Experimental details are given in Sections 4.1 and 4.4.

*Table 2.* **Qualitative example of code watermarking** on a sample from MBPP for which the tests pass (using Llama-3.1-8B at temperature 1.4 with Gumbel-max watermarking, top-$p = 0.95$). The post-hoc rephrasing here refactors variable names to ensure detection (low p-value) while maintaining correctness.

| Original Text | Watermarked Rephrasing |
|---|---|
| ```# A python function to identify non-prime numbers. import math def is_not_prime(n):   result = False   for i in range(2,int(math.sqrt(n)) + 1):     if n % i == 0:       result = True   return result``` | ```# A python function to identify non-prime numbers import math  def is_not_prime(number):   verdict = False   for divisor in range(2, int(math.sqrt(number)) + 1):     if number % divisor == 0:       verdict = True   return verdict``` |
| *p*-val:: 0.12   ●   Tokens: 48 | *p*-val:: $3.29 \times 10^{-3}$   ●   Tokens: 57 |

token level; filtering by token entropy does not expose additional signal, it only reduces the number of scored tokens and thus the available statistical evidence. One other and related property of WaterMax is that it cannot be used for active dataset inference through radioactivity. This is explained in detail in Appendix B.

### 4.6 Post-hoc Code Watermarking

**Experimental details.** We evaluate post-hoc watermarking on codes from HumanEval and MBPP (see in subsection 4.1). First, we measure *functional correctness*: the fraction of watermarked code that passes the tests (pass@1). Second, we measure the *detection power*: the true positive rate (TPR) at a fixed false positive rate (FPR), and/or the distribution of $\log_{10}$ *p*-values. These can be computed among all rephrased code, or more interestingly, among only the ones that passed the tests. In practice, the latter is a better indicator of watermark power, since codes that do not pass tests often exhibit degenerative patterns (repeated code or text at the end) which artificially increase the TPR.

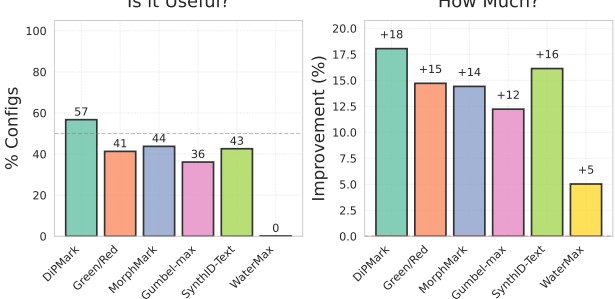

*Figure 5.* **Effect of entropy-aware detection.** *Left:* Share of configurations for which some threshold improves detection by more than $5\%$ on $\geq 50\%$ of the texts. *Middle:* Median improvement for those configurations. Details are given in Sections 4.1 and 4.5.

**Comparing watermarking methods.** We compare the watermarking methods using Llama-3.1-8B. Figure 6 shows the trade-off between functional correctness (pass@1) and detection power (TPR at FPR=$10^{-3}$ among correct codes) for each method. Gumbel-max watermarking achieves the best Pareto frontier, offering the most favorable trade-off between utility and detectability. SynthID performs well in moderate temperature regimes but breaks down at higher temperatures, which limits its ability to achieve high TPR values. MorphMark and DiPMark show competitive performance at intermediate operating points, while Green-Red and SWEET exhibit steeper degradation in pass@1 as watermark strength increases. Overall, no method achieves both high pass@1 and high TPR, confirming the inherent difficulty of watermarking code while preserving functionality.

**Comparing models.** We first compare Llama models of different sizes using Gumbel-max (top-$p$=0.95). The distribution of $p$-values across model families is shown in Figure 10 of the appendix. Different families and sizes behave differently in terms of detection power at fixed temperatures. Table 3 shows that Llama-70B achieves median -$\log_{10}$ p-values around 0.5-0.7, while smaller models reach values

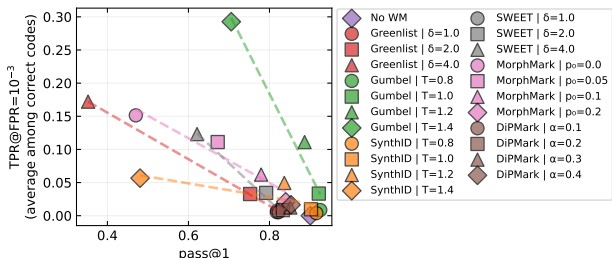

*Figure 6.* **Comparison of methods on code** using Llama-3.1-8B. We report pass@1 (functional correctness) vs. TPR at FPR=$10^{-3}$ among correct codes, averaged over HumanEval and MBPP. Different markers and colors correspond to different watermarking methods and their respective hyperparameters. Consistent with our findings in literary English, Gumbel-max watermarking achieves the best trade-off between utility and detectability.

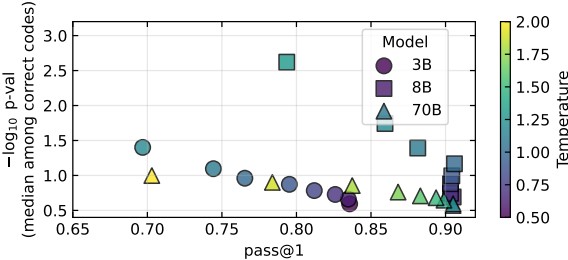

*Figure 7.* **Evaluation of post-hoc watermarking across Llama models** using Gumbel-max (top-$p$=0.95). Pass@1 and detection power (median -$\log_{10}$ p-value among correct codes) for Llama 3B, 8B and 70B models across temperatures. Interestingly, smaller models can achieve better Pareto optimality than larger models.

*Table 3.* **Post-hoc watermarking on code with Llama models** using Gumbel-max (top-$p$=0.95). We report pass@1, median $-\log_{10}$ $p$-value (↑), and TPR at FPR=$10^{-3}$ for different model sizes and temperatures ($T$). Metrics are shown over all samples ("All") and restricted to samples where the watermarked code passes functional tests ("Passed"). Increasing model size increases pass@1 but decreases detection power, with smaller models achieving better Pareto optimality.

| Model | $T = 1.0$ | | | | | $T = 1.2$ | | | | | $T = 1.4$ | | | | |
| | pass@1 | -$\log_{10}$ p-val | | TPR@$10^{-3}$ | | pass@1 | -$\log_{10}$ p-val | | TPR@$10^{-3}$ | | pass@1 | -$\log_{10}$ p-val | | TPR@$10^{-3}$ | |
| | | All | Passed | All | Passed | | All | Passed | All | Passed | | All | Passed | All | Passed |
|---|---|---|---|---|---|---|---|---|---|---|---|---|---|---|---|
| Llama-3.2-1B-Instruct | 0.59 | 0.60 | 0.51 | 0.03 | 0.01 | 0.44 | 1.32 | 0.88 | 0.22 | 0.07 | 0.16 | 9.10 | 1.56 | 0.72 | 0.29 |
| Llama-3.2-3B-Instruct | 0.81 | 0.60 | 0.56 | 0.01 | 0.01 | 0.73 | 0.87 | 0.75 | 0.08 | 0.05 | 0.53 | 1.84 | 1.00 | 0.34 | 0.13 |
| Llama-3.1-8B-Instruct | 0.92 | 0.74 | 0.74 | 0.04 | 0.03 | 0.89 | 1.16 | 1.11 | 0.12 | 0.11 | 0.71 | 2.31 | 1.83 | 0.41 | 0.29 |
| Llama-3.3-70B-Instruct | 0.92 | 0.38 | 0.37 | 0.00 | 0.00 | 0.92 | 0.41 | 0.40 | 0.00 | 0.00 | 0.90 | 0.44 | 0.44 | 0.00 | 0.00 |

near 2 at $T = 1.4$. However, higher detection power in smaller models comes at the cost of functional correctness; for example, at $T = 1.4$, Llama-8B achieves a TPR of 0.29 among correct codes but a pass@1 of only 0.71.

To further investigate this trade-off, we sweep the temperature in Figure 7. Interestingly, the smaller 8B model achieves a better Pareto frontier than the larger 70B model, attaining higher detection power at equivalent pass@1 levels. This suggests that while larger models produce higher-quality code, their reduced entropy limits watermark effectiveness, making smaller models more suitable for post-hoc code watermarking when balancing functionality and detectability. These differences may stem from training conditions (e.g., data, losses) rather than model size alone. Notably, Gemma models exhibit consistently low detection power among correct codes; we hypothesize this stems from RL fine-tuning collapsing the output distribution's entropy.

### 4.7 Multi-Lingual Robustness on Wikipedia

To evaluate post-hoc watermarking beyond English, we apply the method to paragraphs of Wikipedia articles in multiple languages. In the following, we use Llama-3.1-8B-Instruct, with Gumbel-max watermarking with varying temperature settings (0.8, 1.0, 1.2) and fixed top-$p = 0.95$. Figure 8 compares the performance across English (en), Spanish (es), French (fr), and Russian (ru). To not conflate the effect of the number of tokens used for scoring with language, we only consider outputs that contain between

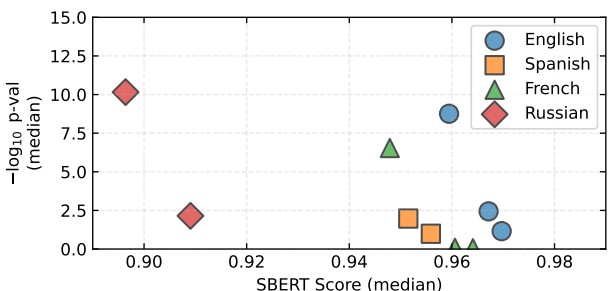

*Figure 8.* **Multi-lingual watermarking performance.** Detection power ($-\log_{10}$ $p$-value) vs. SBERT across languages using Llama-3.1-8B-Instruct with Gumbel-max watermarking. Non-English languages show a steeper quality-detectability trade-off.

450 and 550 tokens, and aggregate results over 1k samples.

The results indicate a performance gap: while non-English languages can be watermarked, they suffer from a steeper trade-off. For Russian for instance, the model occasionally generated invented words at temperatures for which the English rephrasings are perfect. Achieving high detection power in these languages requires a larger sacrifice in semantic quality compared to English. This suggests that the logits of the paraphrasing model are less easy to manipulate in languages for which the model was not primarily trained, since most training data is in English. It may also reflect the fact that semantic similarity metrics remain easier to calibrate in English, although we use the multilingual `paraphrase-multilingual-MiniLM-L12-v2` model for this section.

### 4.8 Impact of Chunking Strategy on Long Documents

To assess the impact of chunking as described in section 3, we compare full-context processing with context-aware chunking (500-token chunks, with up to 1000 context tokens from previously rephrased chunks) on documents of varying length (500-4000 tokens), using Dickens novel excerpts and Llama-3.2-3B-Instruct.

For each document length, results are averaged over 5 independent texts. Table 4 shows that full-context processing increasingly summarizes the content, while chunking better preserves length and yields stronger detection at comparable semantic similarity. For this long-document analysis, we report sentence-level SBERT F1: we split both documents into sentences, encode them independently, and compute precision/recall from the pairwise cosine similarities, avoiding the truncation of fixed-context SBERT encoders. This indicates that context-aware chunking is crucial for reliable watermarking of important document lengths.

## 5 Conclusion and Limitation

We presented a comprehensive evaluation of post-hoc text watermarking via LLM rephrasing. Our experiments yielded several key findings: (1) the Gumbel-max scheme (Aaronson & Kirchner, 2023) achieved better trade-offs than other

*Table 4.* **Context-aware chunking vs. full-context.** Comparison of full-context and context-aware chunked rephrasing (500-token chunks with up to 1000 context tokens) on documents of varying length, averaged over 5 excerpts per length. Experimental details are given in subsection 4.1 and subsection 4.8.

| Size | Length Ratio | | $-\log p$ | | SBERT-F1 | |
|---|---|---|---|---|---|---|
| | *Full* | *Chunked* | *Full* | *Chunked* | *Full* | *Chunked* |
| 500 | 0.91 | 0.91 | 19.3 | 19.3 | 0.80 | 0.80 |
| 1500 | 1.24 | 1.04 | 49.5 | 148.8 | 0.72 | 0.71 |
| 2500 | 0.80 | 1.04 | 56.8 | 195.5 | 0.82 | 0.70 |
| 4000 | 0.68 | 0.93 | 24.9 | 198.3 | 0.63 | 0.68 |

methods under random sampling, and other schemes benefited substantially from beam search; (2) smaller, more entropic models outperformed larger models at high temperature for achieving strong watermarks; (3) entropy filtering at detection provided only marginal gains. Code will be open-sourced to facilitate research in the field.

A limitation of our study is the reliance on automated metrics such as cross entropy and SBERT to approximate quality and semantic preservation, though an additional LLM-as-a-judge validation in subsection B.4 found no significant quality differences among the main watermarking methods. To address this, we extended our analysis to code watermarking, where execution-based correctness provides an objective utility measure. Our results show that while watermarking is feasible in this domain, strict correctness constraints reduce the available capacity for watermarking compared to natural language.

## Impact Statement

This paper presents work whose goal is to advance the field of Machine Learning, specifically in the area of AI-generated content traceability. While watermarking techniques can help identify AI-generated text and protect copyrighted materials, they could also be misused in surveillance contexts. We believe the benefits of transparency and copyright protection outweigh these risks.

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

# A    Additional Details on LLM Watermarking

## A.1    Hash Function

All watermarking schemes in this work rely on a pseudorandom function (PRF) to deterministically map how watermark windows should influence the next-token selection (e.g., green/red classification or Gumbel noise values). The PRF takes as input the candidate token $x$, a context window $\mathbf{w} = (w_1, \ldots, w_k)$ of $k$ token IDs, and the secret key $s$ (all of them are integers), and outputs a random integer in $[0, M)$. We compute:

$$h'(x, \mathbf{w}, s) = \left( p_2 \cdot x + \sum_{i=1}^{k} w_i \cdot q_i + p_3 \cdot s \right) \cdot p_4, \tag{1}$$

$$h(x, \mathbf{w}, s) = \text{XORShift}(h'(x, \mathbf{w}, s)) \mod M, \tag{2}$$

where $q_1, \ldots, q_k$ are distinct large primes (to ensure that different orderings of the same tokens produce different tokens), and $p_2, p_3, p_4$ are additional primes. The first result $h'$ undergoes XOR-shift for better bit dispersion: $h = (h' \cdot p_{\text{mix}}) \oplus ((h' \cdot p_{\text{mix}}) \gg s)$, where $p_{\text{mix}}$ is a mixing prime and $s$ is a shift constant. Finally, $h \mod M$ yields an integer in $[0, M)$.

For uniform PRF output in $[0, 1)$, we divide by $M$. For binary (green/red) classification, we threshold at $\gamma$: a token is "green" if $h/M < \gamma$. This construction ensures that small changes in any input (window, token, or key) produce uncorrelated outputs, satisfying the pseudo-randomness requirements for watermark security. In practice, the implementation allows for selectively scoring some tokens without needing to generate the full green/red list, which can speed up computations for large vocabularies, especially for SynthID-text, which uses multiple lists for different $g$-values.

## A.2    Details on Watermark Schemes

We describe below the main watermarking schemes we evaluate.

**Green-list/red-list.**    Kirchenbauer et al. (2023a;b) modify the logit vector during next-token generation based on the watermark window of $k$ previous tokens and the private key $s$. Note that the case where $k = 0$ corresponds to the work of Zhao et al. (2023), but we found that it empirically breaks theoretical assumptions and often leads to degenerate text; therefore, we use $k \geq 1$ in our experiments. For each token $v \in \mathcal{V}$, the pseudorandom function $\text{PRF}(\cdot)$ outputs a value in $[0, 1)$ from the token ID $v$, the context window, and $s$. A token is classified as "green" if $\text{PRF}(v, \mathbf{w}, s) < \gamma$, where $\gamma \in [0, 1]$ is the expected proportion of green tokens under the null hypothesis (typically $\gamma = 0.5$). Logits of green tokens are incremented by $\delta$ to increase their sampling probability:

$$\tilde{\ell}_v = \begin{cases} \ell_v + \delta & \text{if } v \in \text{GreenList} \\ \ell_v & \text{otherwise} \end{cases} \tag{3}$$

Detection involves repeating the greenlist computation for each token of a text, incrementing a score by 1 if the token is in the greenlist, and performing a statistical test on the cumulative score. Under the null hypothesis $\mathcal{H}_0$ "the text is not watermarked with that scheme and private key $s$", this score follows a binomial distribution. A simple binomial test thus provides a $p$-value: the probability of observing at least as many green tokens as observed, if the text was not watermarked.

**SWEET.**    Lee et al. (2023) apply the Green-Red list watermark only to high-entropy tokens. The intuition is that low-entropy tokens (where the model is confident) contribute little signal but can degrade quality if biased. At generation time, tokens with entropy below a threshold are sampled without watermark bias. Specifically, a token is considered high-entropy if $H(\mathbf{p}) = -\sum_{v \in \mathcal{V}} p_v \log p_v > \tau$, where $\tau$ is a predefined threshold. At detection time, these tokens are similarly excluded from scoring, improving signal-to-noise ratio. Note that this filtering can also be applied only at detection time, and on top of other schemes than green-red (as we do in our experiments).

**MorphMark.**    Wang et al. (2025) adaptively adjust watermark strength based on context. Let $P_G = \sum_{v \in \text{GreenList}} p_v$ be the total probability mass on green tokens before watermarking. If $P_G \leq p_0$ (a threshold, e.g., $p_0 = 0.15$), no watermark is applied to preserve quality. Otherwise, an adaptive boost factor $r = \min(\kappa P_G, 1)$ is computed, where $\kappa$ controls the

watermark strength. Probabilities are then adjusted:

$$\hat{p}_v = \begin{cases} p_v \cdot \left(1 + \frac{r(1-P_G)}{P_G}\right) & \text{if } v \in \text{GreenList} \\ p_v \cdot (1 - r) & \text{otherwise} \end{cases} \tag{4}$$

This ensures stronger watermarking when the green list is already favorable, minimizing quality degradation.

**DiPMark.** Wu et al. (2023) introduce a variant of Green-Red watermarks that is distortion-free. The method uses a pseudorandom permutation $\pi$ (seeded by the context window and $s$) to reorder tokens. After permutation, the cumulative probability distribution is modified as follows: tokens in the interval $[0, 1 - \alpha]$ have their probability set to zero, tokens in $[1 - \alpha, \alpha]$ remain unchanged, and tokens in $[\alpha, 1]$ have their probability doubled (then renormalized). This creates a detectable bias while preserving the original distribution's average over the randomness of $\pi$.

**Gumbel-max.** Aaronson & Kirchner (2023) alternatively leverage the "Gumbel-max trick". After applying temperature scaling and optional top-$k$ or top-$p$ filtering to obtain a probability vector $\mathbf{p}$, the next token is selected as:

$$x_t = \arg\max_{v \in \mathcal{V}} r_v^{1/p_v} \tag{5}$$

where $r_v \sim \text{Uniform}(0, 1)$ is i.i.d. noise (which is equivalent to sampling from $\mathbf{p}$). The watermark intervenes by replacing the purely random $r_v$ with pseudorandom values $r_v = \text{PRF}(v, \mathbf{w}, s) \in [0, 1)$ generated by the PRF from the token ID $v$, the context window, and $s$, for each $v \in \mathcal{V}$. Consequently, for a fixed context, the noise vector is deterministic. Detection is performed by recomputing the PRF output $r_{x_t} = \text{PRF}(x_t, \mathbf{w}_t, s)$ for each observed token $x_t$ and computing the cumulative score:

$$S = \sum_{t=1}^{N} -\log(1 - r_{x_t}) \tag{6}$$

where $N$ is the number of tokens in the text. Under $\mathcal{H}_0$ "the text is not watermarked with that scheme and private key $s$", $S$ follows a $\Gamma(N, 1)$ distribution. The $p$-value of a test associated with score $S$ reads: $p\text{-value}(S) = \Gamma(N, S)/\Gamma(N)$, where $\Gamma$ is the upper incomplete gamma function.

**SynthID-text.** Dathathri et al. (2024) propose SynthID Text, which employs a tournament-based sampling strategy. The method uses $m$ scoring functions $g_1, \ldots, g_m$, where each $g_i$ maps a token to a pseudorandom value in $\{0, 1\}$ based on the context window and secret key[1]. Unlike the additive bias of the "Green-Red" list method, this approach samples $2^m$ candidate tokens from the model's distribution and randomly organizes them into a tournament where winners are determined by the $g$-values. Detection relies on a statistical hypothesis test using the mean $g$-value of the observed tokens as the test statistic. Under the null hypothesis $\mathcal{H}_0$ (unwatermarked text), the $g$-values are independent and identically distributed (e.g., uniform or Bernoulli). This scheme is shown to outperform Green-Red approaches because the tournament depth $m$ allows for a more favorable trade-off between detectability and text quality (perplexity), particularly in low-entropy settings where standard additive biases often degrade coherence.

Scoring Functions: Mean vs. Weighted Mean. While the standard detection method computes a simple unweighted mean of the $g$-values across all tokens and tournament layers (effectively a normalized sum analogous to a binomial test), Dathathri et al. (2024) demonstrate that this is theoretically suboptimal for multi-layer tournaments. The authors observe that the amount of "watermarking evidence" embedded by the tournament is not uniform across layers; specifically, the strength of the watermark diminishes as the tournament depth increases because each successive layer operates on a subset of candidates with reduced entropy. Consequently, an unweighted mean dilutes the strong signal from early layers with the weaker signal from deeper layers. To address this, SynthID-text utilizes a *weighted mean score*, which assigns decreasing weights $w_l$ to the $g$-values of the $l$-th tournament layer. By emphasizing the earlier layers where the statistical signature is most robust, the weighted scoring function improves the signal-to-noise ratio of the test statistic, yielding higher detection accuracy (true positive rate) for a fixed false positive rate compared to the classical unweighted approach.

---

[1]In our implementation, each $g$-function corresponds to a distinct green/red list partition: we compute them by incrementing the token ID by $i$ for each depth $i$ of the tournament. Since the hash function exhibits no correlation for incremented token IDs, this effectively simulates independent green/red lists per tournament layer.

## A.3 Details on Radioactivity

**The Radioactivity Test Protocol.** To formally test for watermarked dataset radioactivity, we detail the methodology from Sander et al. (2024; 2025) for the Green-list/Red-list scheme. The core idea is to repurpose the standard watermark detection test (normally applied to observed text tokens) and instead apply it to the *predicted* tokens of the suspect model. This allows us to determine whether the model is familiar with the watermark, thereby providing evidence that the model was exposed to the watermark during training.

**Teacher Forcing Setup.** We feed the *watermarked* text $\mathbf{y}$ into the suspect model $f_\theta$ using a teacher-forcing setup. Let $\hat{y}_t$ denote the top-1 prediction of the suspect model at step $t$, given the watermarked prefix $y_{<t}$:

$$\hat{y}_t = \underset{v \in \mathcal{V}}{\operatorname{argmax}}\, P_\theta(v \mid y_{<t}). \tag{7}$$

**Test Statistic.** We define the radioactivity score $S$ as the empirical proportion of the suspect model's predictions that fall into the Green-list:

$$S = \frac{1}{|\mathcal{U}|} \sum_{t \in \mathcal{U}} \mathbb{I}\big(h(\hat{y}_t, y_{t-k:t-1}, s) < \gamma\big), \tag{8}$$

where $s$ is the secret key and $\gamma$ is the expected green ratio (typically $0.5$).

**Deduplication.** To ensure statistical independence, we must account for repeated n-grams in the context that could influence the model to reproduce them. In particular, if an n-gram in context is green, the suspect model might repeat it due to context copying rather than watermark radioactivity. We filter the indices $t$ to form a set $\mathcal{U}$ such that each watermark window $y_{t-k:t-1}$ is only scored once, which fully removes this issue (Sander et al., 2024). Under the null hypothesis $\mathcal{H}_0$ (i.e., the suspect model is unaware of $s$), the count of green predictions should now follow a binomial distribution, as the suspect model should exhibit no preference toward green or red tokens:

$$K \sim \operatorname{Binomial}(|\mathcal{U}|, \gamma). \tag{9}$$

This formulation allows for the computation of an exact $p$-value.

**WaterMax Is Not Radioactive.** For the score to truly follow a known binomial distribution under $\mathcal{H}_0$ ("the suspect model has never been in contact with the watermark"), the watermark bias must be applied at the *token level*. However, WaterMax selects the final sequence $x^*$ from a set of candidates $\mathcal{C}$ by maximizing the number of green tokens globally *by chance*. Consequently, an innocent model $f_\theta$ that shares a similar language distribution with the generator $P_{\text{gen}}$ will find that its optimal next token $\hat{y}_t$ is green significantly more often than $\gamma$, purely due to this selection bias.

We verified this experimentally by running the radioactivity detection test on approximately 100k watermarked tokens generated via WaterMax, Green-list/Red-list random sampling, and Green-list/Red-list beam search with biased scoring (after rephrasing 300 excerpts from Dickens as detailed in subsection 4.1). As expected, the radioactivity test yields a $p$-value of 0.93 for Green-list/Red-list random sampling and 0.76 for Green-list/Red-list beam search with biased scoring, *but* $1.0 \times 10^{-6}$ *for WaterMax*.

# B Additional Experimental Details and Results

## B.1 Choice of the Secret Key and Statistical Correctness

**How to choose the secret key?** As explained in section 3, valid detection requires $p$-values to be uniformly distributed under $\mathcal{H}_0$. While this holds in expectation over all possible keys, fixing a specific key $s$ creates preferences for certain $(k+1)$-grams. If these patterns align with natural language statistics, the expected green fraction under $\mathcal{H}_0$ shifts slightly (e.g., from 0.5 to 0.505), causing $p$-values to collapse toward zero on long texts and inflating false positive rates.

We address this by testing 50 candidate keys for each combination of tokenizer, watermarking method, and relevant hyperparameters. Each key is evaluated on 100 non-watermarked texts of 800 characters, and we select the key that minimizes deviation from $U(0,1)$ as measured by the Kolmogorov-Smirnov (KS) statistic. This ensures our results reflect intrinsic scheme performance rather than key-specific artifacts. Detailed statistics are provided in Table 5.

**Empirical validation of false positive rates.** We verify that the selected keys yield valid statistical tests by computing scores and $p$-values on 1.5M Wikipedia documents (averaging 715 tokens each). For each watermarking method and its best key (as presented in Table 5), we compute the empirical FPR at various thresholds and confirm they match the theoretical values. We perform this validation both with deduplication using the watermark window ($k$-gram) and with deduplication using the watermark window plus the scored token (($k + 1$)-gram). Figure 9 shows that empirical FPRs closely track theoretical expectations across all methods.

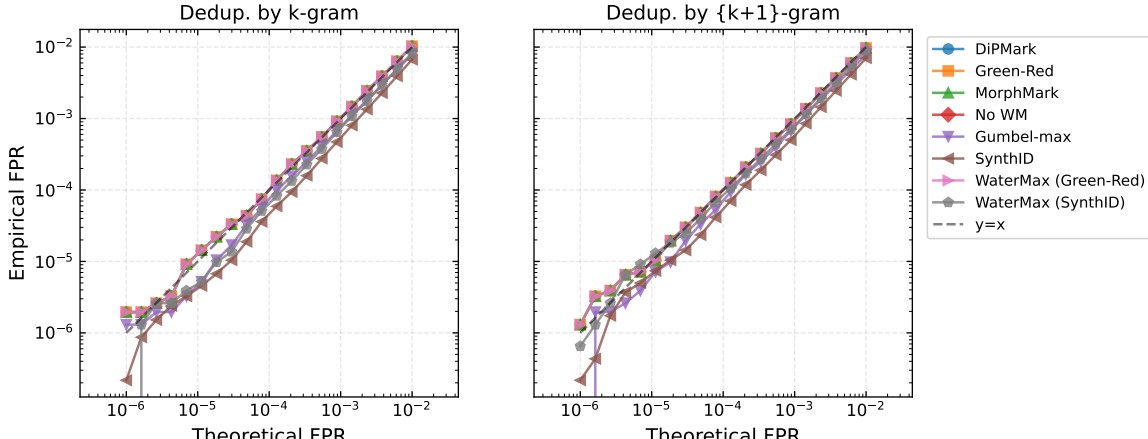

*Figure 9.* **Empirical vs. theoretical FPR validation.** Comparison of expected and observed false positive rates on Wikipedia documents confirms that the selected keys yield statistically valid tests.

## B.2 More Details on Code Experiments

This section provides additional details on the code watermarking experiments from subsection 4.6.

**Data preparation.** For each HumanEval and MBPP problem, we concatenate the provided prompt (function signature and docstring) with the canonical solution. We ensure the last top-level function definition is the target for execution, moving it if necessary. An example is shown in Figure 11.

**Postprocessing.** Since watermarking may inadvertently rename function identifiers, we implement a postprocessing step that parses the output using Python's AST module and restores the original entry point name required by the benchmark test harness. At evaluation time, we concatenate the watermarked code with the original test harness to assess functional correctness (Pass@1).

**Distribution of detection power across models.** Figure 10 shows the distribution of $-\log_{10}$ p-values for watermarked codes that pass the functional correctness test across different model families. Different colors represent model families, with darker shades indicating larger models. Larger models lead to lower detection power (lower $-\log_{10}$ p-values), as their outputs are less entropic and leave less room for watermarking.

*Table 5.* Key sensitivity analysis across models and watermarking schemes. We evaluate multiple random keys per configuration on 100 non-watermarked texts. "SynthID (W)" denotes the Weighted variant which utilizes a different scoring statistic. The **Best Key** is selected based on the highest $p$-value from the Kolmogorov-Smirnov (KS) test for uniformity under $\mathcal{H}_0$.

| Model | Method | Aggregated Statistics (All Keys) | | Selected "Best" Key | | |
|---|---|---|---|---|---|---|
| | | Avg. $p$-val | $\sigma_{\textbf{keys}}$ | Key ID | Avg. $p$-val | KS $p$ |
| **Qwen 2.5** | DipMark | 0.501 | 0.033 | #596061 | 0.493 | 0.99 |
| | Green-Red | 0.501 | 0.033 | #596061 | 0.493 | 0.99 |
| | MorphMark | 0.501 | 0.033 | #596061 | 0.493 | 0.99 |
| | Gumbel-max | 0.497 | 0.031 | #2345 | 0.499 | 0.99 |
| | SynthID ($d = 10$) | 0.498 | 0.035 | #606 | 0.494 | 0.95 |
| | SynthID ($d = 20$) | 0.488 | 0.033 | #753 | 0.506 | 0.99 |
| | SynthID ($d = 30$) | 0.488 | 0.031 | #1357 | 0.493 | 0.97 |
| | SynthID (W) ($d = 10$) | 0.498 | 0.030 | #323334 | 0.506 | 0.98 |
| | SynthID (W) ($d = 20$) | 0.493 | 0.029 | #3452 | 0.498 | 0.99 |
| | SynthID (W) ($d = 30$) | 0.490 | 0.032 | #505152 | 0.509 | 1.00 |
| **Llama 3** | DipMark | 0.508 | 0.037 | #323334 | 0.505 | 0.95 |
| | Green-Red | 0.508 | 0.037 | #323334 | 0.505 | 0.95 |
| | MorphMark | 0.508 | 0.037 | #323334 | 0.505 | 0.95 |
| | Gumbel-max | 0.498 | 0.030 | #6780 | 0.506 | 1.00 |
| | SynthID ($d = 10$) | 0.507 | 0.035 | #656667 | 0.504 | 0.99 |
| | SynthID ($d = 20$) | 0.497 | 0.035 | #626364 | 0.493 | 0.99 |
| | SynthID ($d = 30$) | 0.503 | 0.037 | #686970 | 0.499 | 0.99 |
| | SynthID (W) ($d = 10$) | 0.504 | 0.036 | #888 | 0.507 | 0.96 |
| | SynthID (W) ($d = 20$) | 0.500 | 0.034 | #42 | 0.504 | 1.00 |
| | SynthID (W) ($d = 30$) | 0.502 | 0.036 | #333 | 0.490 | 0.98 |
| **Gemma 3** | DipMark | 0.501 | 0.035 | #8907 | 0.488 | 0.96 |
| | Green-Red | 0.501 | 0.035 | #8907 | 0.488 | 0.96 |
| | MorphMark | 0.501 | 0.035 | #8907 | 0.488 | 0.96 |
| | Gumbel-max | 0.498 | 0.034 | #6789 | 0.506 | 0.97 |
| | SynthID ($d = 10$) | 0.503 | 0.041 | #656667 | 0.504 | 0.98 |
| | SynthID ($d = 20$) | 0.496 | 0.036 | #323334 | 0.488 | 0.99 |
| | SynthID ($d = 30$) | 0.502 | 0.036 | #131415 | 0.495 | 0.93 |
| | SynthID (W) ($d = 10$) | 0.501 | 0.042 | #1234 | 0.519 | 0.97 |
| | SynthID (W) ($d = 20$) | 0.492 | 0.036 | #258 | 0.501 | 0.99 |
| | SynthID (W) ($d = 30$) | 0.503 | 0.039 | #222324 | 0.498 | 0.98 |
| **SmolLM2** | DipMark | 0.496 | 0.034 | #789 | 0.506 | 1.00 |
| | Green-Red | 0.496 | 0.034 | #789 | 0.506 | 1.00 |
| | MorphMark | 0.496 | 0.034 | #789 | 0.506 | 1.00 |
| | Gumbel-max | 0.507 | 0.038 | #252627 | 0.498 | 1.00 |
| | SynthID ($d = 10$) | 0.506 | 0.044 | #535455 | 0.501 | 0.98 |
| | SynthID ($d = 20$) | 0.495 | 0.041 | #369 | 0.499 | 1.00 |
| | SynthID ($d = 30$) | 0.497 | 0.043 | #444 | 0.500 | 0.99 |
| | SynthID (W) ($d = 10$) | 0.502 | 0.043 | #707 | 0.509 | 0.98 |
| | SynthID (W) ($d = 20$) | 0.499 | 0.041 | #369 | 0.499 | 1.00 |
| | SynthID (W) ($d = 30$) | 0.501 | 0.044 | #252627 | 0.507 | 0.98 |

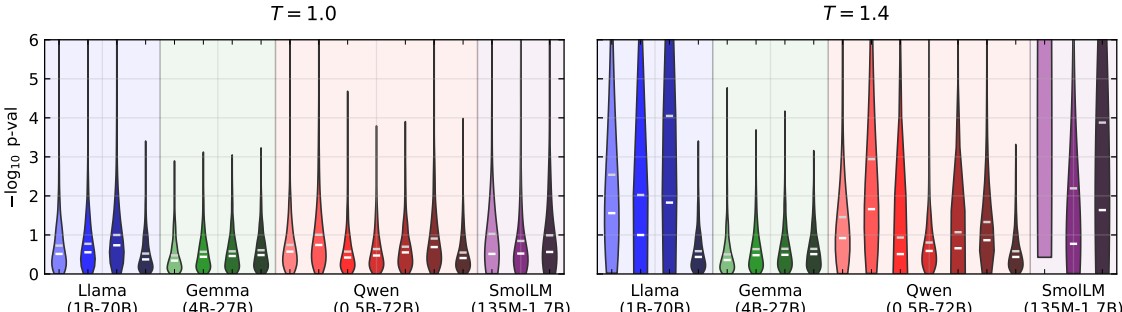

*Figure 10.* **Distribution of detection power across model families** using Gumbel-max (top-$p$=0.95). Violin plots show the distribution of $-\log_{10}$ p-value for watermarked codes that pass the functional correctness test (bars indicate mean and median). Different colors represent model families, with darker shades indicating larger models. Larger models generally achieve lower detection power due to reduced output entropy.

**Input (Prompt + Canonical Solution):**

```python
from typing import List

def has_close_elements(numbers: List[float],
    threshold: float) -> bool:
    """ Check if in given list of numbers, are any
     two numbers closer to each other than given
     threshold.
    >>> has_close_elements([1.0, 2.0, 3.0], 0.5)
    False
    >>> has_close_elements([1.0, 2.0, 5.0, 2.0],
     0.3)
    True
    """
```

```python
    for idx, elem in enumerate(numbers):
        for idx2, elem2 in enumerate(numbers):
            if idx != idx2:
                distance = abs(elem - elem2)
                if distance < threshold:
                    return True
    return False
```

**Test:**

```python
METADATA = { 'author': 'jt', 'dataset': 'test' }

def check(candidate):
    assert candidate([1.0, 2.0, 3.9, 4.0, 5.0, 2],
      0.3) == True
    assert candidate([1.0, 2, 3, 4.0, 5.0, 2.2],
      0.05) == False
    assert candidate([1.0, 2.0, 5.9, 4.0, 5.0,
      0.95]) == True
    assert candidate([1.0, 2.0, 5.9, 4.0, 5.0,
      0.8]) == False
    assert candidate([1.0, 2.0, 3.0, 4.0, 5.0, 2],
      0.1) == True

check(has_close_elements)
```

*Figure 11.* Example of a HumanEval task. The left block represents the text subjected to post-hoc watermarking (paraphrasing). The right block is the test harness used to verify functional correctness (Pass@1).

## B.3  Are the Pareto take-aways the same after attacks?

**Rephrasing Attack Simulation.**  To evaluate the robustness of watermarks against text rephrasing, we simulate attacks by prompting a language model to paraphrase the watermarked text (without any watermark bias). We use `Llama-3.2-3B-Instruct` as the attack model with standard nucleus sampling ($p = 0.95$). We consider three attack intensities with increasing aggressiveness via temperature and system prompt:

- **Very Light** ($T = 0.5$): The model is instructed to act as a "careful text editor" making only minimal changes such as grammar and punctuation fixes while preserving the original wording.

- **Light** ($T = 0.8$): The model is prompted as a "text rephrasing assistant" that rephrases the text while preserving its meaning and style, using different words and sentence structures where appropriate.

- **Medium** ($T = 1.0$): The model is instructed as an "expert paraphrasing tool" to completely rewrite the text using entirely different words, phrases, and sentence structures while keeping the core meaning intact.

The attack uses vanilla autoregressive generation without any watermark logit manipulation, simulating a realistic adversarial scenario. For each attack, we measure watermark detection via $p$-value on the rephrased text and semantic preservation using BERTScore similarity.

To quantify attack strength independently of the watermarking method, we propose a comparative metric that measures attack-induced degradation relative to watermark-induced degradation. We compute

| Attack Level | Attacked Quality | Strength Ratio |
|---|---|---|
| Very Light | 0.932 | $0.597 \pm 0.119$ |
| Light | 0.885 | $0.863 \pm 0.215$ |
| Medium | 0.860 | $1.095 \pm 0.252$ |

*Table 6.* **Attack strength metrics.** Attacked quality shows BERTScore between attacked and watermarked text. Strength ratios quantify attack severity relative to watermark degradation.

BERTScore between (1) watermarked and original text ($\text{SBERT}_{\text{wm}}$), and (2) attacked and watermarked text ($\text{SBERT}_{\text{attacked}}$). The *attack strength ratio* is defined as $(1 - \text{SBERT}_{\text{attacked}})/(1 - \text{SBERT}_{\text{wm}})$, representing how much quality loss the attack introduces relative to the watermark's quality loss. A ratio of 1.0 indicates the attack degrades the watermarked text by the same amount as the watermark did on the original text; ratios below 1.0 indicate weaker attacks that preserve more similarity than the watermark; ratios above 1.0 indicate stronger attacks. (Median) results are provided in Table 6, across all methods and set-ups (the median SBERT between watermarked and original was 0.878).

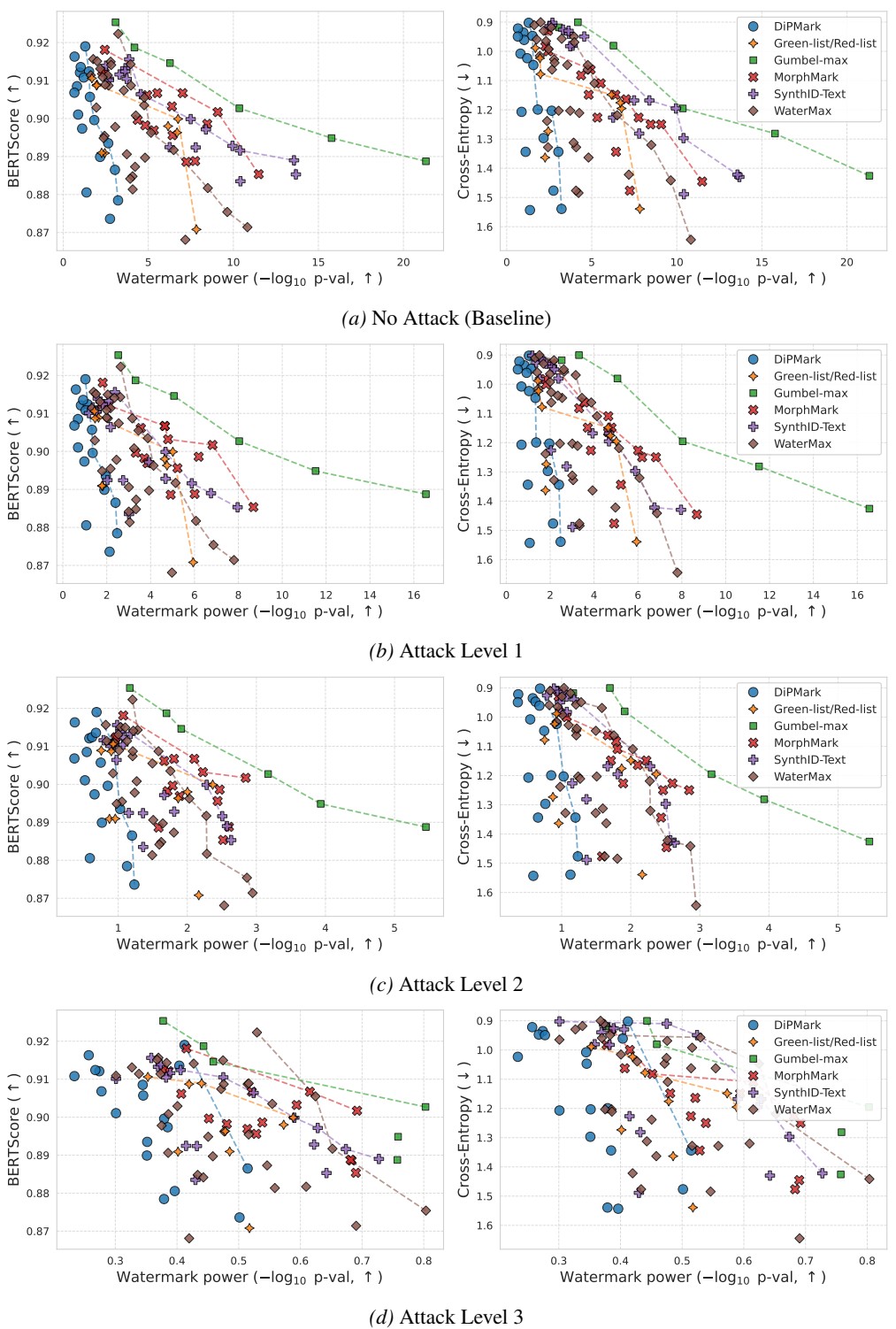

*(a)* No Attack (Baseline)

*(b)* Attack Level 1

*(c)* Attack Level 2

*(d)* Attack Level 3

*Figure 12.* **Attack strength analysis.** (a) Baseline Pareto frontier showing the trade-off between watermark strength and text quality across different watermarking methods without attacks. (b–d) Quality preservation at different attack levels. Each point is one experimental configuration. X-axis: BERTScore(WM, Original); Y-axis: BERTScore(Attacked, WM). Experimental details are given in subsection 4.2 and Appendix B. Gumbel-max consistently outperform other methods.

## B.4 Additional Camera-Ready Experiments

We report here the additional experiments requested during review. These results do not change the main conclusion: Gumbel-max remains the strongest post-hoc watermarking method on the quality-detectability frontier for open-ended text, while stronger rewriting attacks remove token-level watermarks at a measurable quality cost.

**Domain generalization.** Table 7 repeats the main method comparison beyond Dickens on English Wikipedia and Reddit ELI5 passages using Llama-3.2-3B-Instruct. The absolute detection strength varies by domain, with lower signal on factual Wikipedia text, but the relative ranking is stable: Gumbel-max is consistently strongest, MorphMark is usually second, and DiPMark is weakest.

*Table 7.* **Domain generalization.** Median semantic similarity and detection strength for 100 passages per domain.

| Domain | Method | SBERT | $-\log_{10} \mathbf{p}$ |
|---|---|---|---|
| Wikipedia, $T = 0.7$ | Gumbel-max | 0.977 | 1.32 |
| | Green-Red | 0.977 | 0.89 |
| | SynthID | 0.979 | 0.84 |
| | MorphMark | 0.976 | 1.24 |
| | DiPMark | 0.977 | 0.65 |
| ELI5, $T = 0.7$ | Gumbel-max | 0.849 | 2.24 |
| | Green-Red | 0.841 | 1.37 |
| | SynthID | 0.856 | 1.93 |
| | MorphMark | 0.842 | 2.19 |
| | DiPMark | 0.859 | 0.86 |
| Wikipedia, $T = 1.0$ | Gumbel-max | 0.971 | 2.69 |
| | Green-Red | 0.963 | 1.45 |
| | SynthID | 0.967 | 1.70 |
| | MorphMark | 0.961 | 2.29 |
| | DiPMark | 0.968 | 0.84 |
| ELI5, $T = 1.0$ | Gumbel-max | 0.814 | 5.71 |
| | Green-Red | 0.810 | 2.19 |
| | SynthID | 0.827 | 3.19 |
| | MorphMark | 0.814 | 4.18 |
| | DiPMark | 0.816 | 1.46 |

**Prompt sensitivity.** Table 8 evaluates three rephrasing prompts on 100 Dickens passages with Gumbel-max. The minimal prompt gives the strongest watermark because it gives the paraphraser more freedom, but all variants preserve the same qualitative trade-off between rephrasing freedom and watermark capacity.

*Table 8.* **Prompt sensitivity.** Gumbel-max on 100 Dickens passages under three prompt variants.

| Prompt | ROUGE-L | Cross Ent. | $-\log_{10} \mathbf{p}$ |
|---|---|---|---|
| Default | 0.511 | 1.83 | 4.40 |
| Minimal | 0.353 | 1.95 | 6.31 |
| Elaborate | 0.441 | 2.02 | 5.61 |

**Translation and style-transfer attacks.** Table 9 shows that translation attacks remove the token-level watermark nearly as effectively as moderate paraphrasing. This is an expected limitation of token-level statistical watermarks: translation rewrites the token sequence almost completely. Cross-model and style-transfer attacks showed the same qualitative Pareto ordering as the paraphrase attacks in Figure 12.

**Interaction with pre-existing watermarks.** We also tested whether a new post-hoc watermark can be embedded when the input text already contains a generation-time watermark. Using 100 Dickens passages, we first embedded a Green-Red watermark with one key and then rephrased with Gumbel-max under a separate key. Table 10 shows that the new Gumbel-max signal is essentially unchanged by the input's prior watermark status. The residual Green-Red signal after rephrasing is

*Table 9.* **Translation attacks.** Gumbel-max watermarked Dickens passages attacked with Qwen-2.5-3B-Instruct.

| Setting | Median $-\log_{10} \mathbf{p}$ |
|---|---|
| No attack | 4.4 |
| Paraphrase attack, $T = 1.0$ | 0.5 |
| Translation, en→fr→en | 0.8 |
| Translation, en→ru→en | 0.5 |

also comparable with or without applying the new watermark, supporting the expected separation from independent keys. Highly robust semantic watermarks may survive rephrasing, but their presence does not enter the PRF used to bias the post-hoc output distribution.

*Table 10.* **Interaction with pre-existing watermarks.** Detection strength on 100 Dickens passages.

| Condition | $-\log_{10} \mathbf{p}$ |
|---|---|
| Green-Red before rephrasing | 2.90 |
| Gumbel-max on watermarked input | 1.60 |
| Gumbel-max on non-watermarked input | 1.70 |
| Residual Green-Red after Gumbel-max rephrasing | 1.19 |
| Residual Green-Red after unwatermarked rephrasing | 1.13 |

**Compute overhead.** Table 11 reports wall-clock generation time for 100 watermarked passages of roughly 200 tokens on a single H100 GPU. Gumbel-max adds negligible overhead over unwatermarked sampling in our implementation, while beam search and WaterMax are more expensive. These numbers are from a research prototype: Green-Red scores the full vocabulary, Gumbel-max uses compiled kernels over top-$p$ survivors, and WaterMax uses sequential Python loops.

*Table 11.* **Wall-clock overhead.** Llama-3.2-3B-Instruct, 100 passages, single H100 GPU.

| Method | Time (s) | vs. no WM |
|---|---|---|
| No watermark | 304 | 1.00× |
| Gumbel-max | 307 | 1.01× |
| Green-Red | 322 | 1.06× |
| Green-Red + beam, $B = 3$ | 387 | 1.27× |
| Green-Red + beam, $B = 5$ | 417 | 1.37× |
| WaterMax, $L = 8, m = 4$ | 1376 | 4.53× |

**LLM-as-a-judge validation.** Finally, we ran an LLM-as-a-judge validation using Qwen-2.5-32B-Instruct on 100 passages and five watermarking methods with a 1–5 quality scale. All methods scored above 4.0/5, with no significant differences among methods. This supports the automated quality metrics, but the coarse rating scale is less useful for resolving the fine-grained Pareto trade-offs in the main experiments.

### B.5 Analysis of Output Length Distributions

In the main text, we apply a filtering criterion to retain only rephrasings where the output length remains within a factor of 0.75 to 1.25 of the original input length. In this appendix, we analyze the distribution of these length ratios to validate that this filtering does not introduce bias against specific watermarking schemes.

Figure 13 presents the distribution of length ratios (defined as $L_{\text{output}}/L_{\text{input}}$) across all evaluated models and five distinct watermarking methods. The green shaded region indicates the acceptance window used in our main experiments.

We observe two key trends:

1. **Consistency across schemes:** For any given base model (rows), the distribution of output lengths is remarkably consistent across all watermarking methods (columns). Whether using simple rejection sampling (WaterMax) or distribution perturbation (DiPMark, Gumbel-max), the variance in output length remains stable.

2. **Dependence on model capability:** The ability to respect the length constraint is primarily a function of the model size and instruction-following capability. Smaller models (e.g., SmolLM2-135M) exhibit high variance and frequently generate outputs that are too short or too long, whereas larger, more capable models (e.g., Llama-3.1-8B, Gemma-2-27B) produce tight distributions centered near the ideal ratio of 1.0.

These findings confirm that outliers in length are attributable to the underlying model's generation stability rather than the watermarking process itself. Consequently, filtering these outliers allows for a fairer assessment of semantic preservation and detection power on valid generations, rather than penalizing watermarking schemes for the base model's verbosity or brevity failures.

## C  Additional Text Samples

### C.1  Example of Watermarked Codes

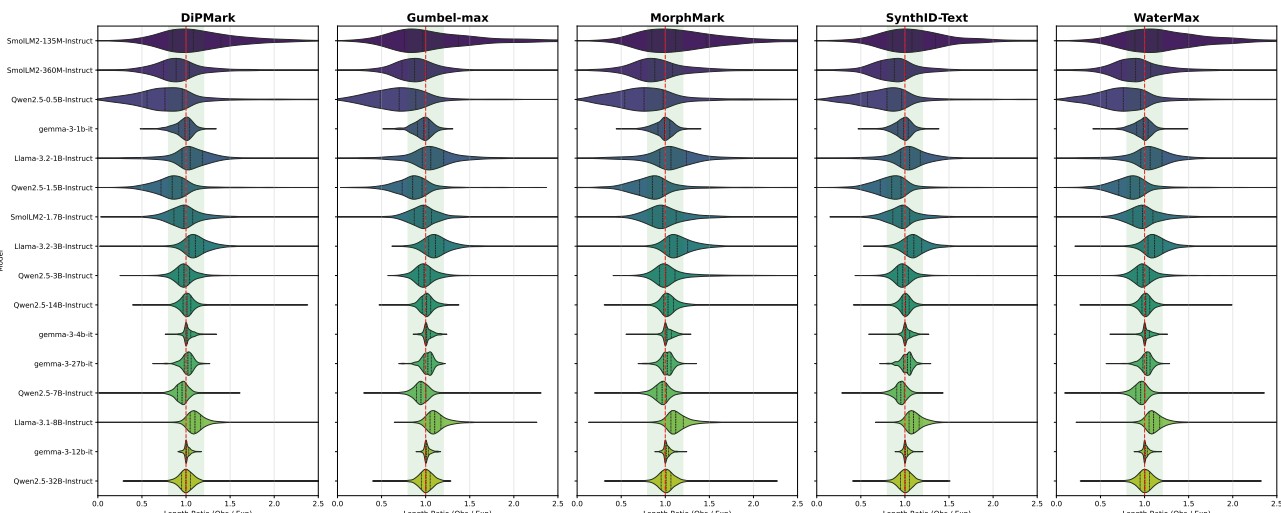

*Figure 13.* **Impact of model choice on output length consistency.** Violin plots showing the distribution of length ratios ($\text{Length}_{wm}/\text{Length}_{orig}$) for varying models and watermarking schemes. The green shaded region represents the inclusion criteria ($[0.75, 1.25]$) used in the main experiments. The red dashed line indicates the ideal ratio of 1.0. Note that length variance is driven primarily by the base model choice (rows) rather than the watermarking scheme (columns), with larger models consistently adhering closer to the target length.

*Table 12.* Post-hoc code watermarking examples on a sample from MBPP (using Llama-3.1-8B at temperatures 0.8, 1.2 and 1.4 with Gumbel-max watermarking, top-$p = 0.95$).

| Original Code | Watermarked Code |
|---|---|

```
# A function to find the minimum cost path to reach (m,
n) from (0, 0) ...
R = 3
C = 3
def min_cost(cost, m, n):
  tc = [[0 for x in range(C)] for x in range(R)]
  tc[0][0] = cost[0][0]
  for i in range(1, m+1):
    tc[i][0] = tc[i-1][0] + cost[i][0]
  for j in range(1, n+1):
    tc[0][j] = tc[0][j-1] + cost[0][j]
  for i in range(1, m+1):
    for j in range(1, n+1):
      tc[i][j] = min(tc[i-1][j-1], tc[i-1][j],
tc[i][j-1]) + cost[i][j]
  return tc[m][n]
```

```
# A function to calculate the minimum cost path from the
origin (0, 0) ...
R = 3
C = 3

def min_cost_path(cost, m, n):
  # Create a temporary cost matrix tc of the same
dimensions...
  tc = [[0 for x in range(C)] for x in range(R)]
  # The cost to reach the first cell (0, 0) is the
cost...
  tc[0][0] = cost[0][0]
  # Calculate the cost to reach the first row.
  for i in range(1, m+1):
    # The cost to reach the first cell in the current
row...
    tc[i][0] = tc[i-1][0] + cost[i][0]
  # Calculate the cost to reach the first column.
  for j in range(1, n+1):
    # The cost to reach the first cell in the current
column...
    tc[0][j] = tc[0][j-1] + cost[0][j]
  # Calculate the cost to reach the rest of the cells.
  for i in range(1, m+1):
    for j in range(1, n+1):
      # The cost to reach the current cell is the
minimum...
      tc[i][j] = min(tc[i-1][j-1], tc[i-1][j],
tc[i][j-1]) + cost[i][j]
  # Return the cost to reach the cell at position (m,
n).
  return tc[m][n]
```

**Stats:** $T$: 0.8 • $p_{\text{value}}$: $2.11e - 01$ • Tokens: 414 • Code correctness: True

```
# A function to find the minimum cost path to reach (m,
n) from (0, 0) ...
R = 3
C = 3
def min_cost(cost, m, n):
  tc = [[0 for x in range(C)] for x in range(R)]
  tc[0][0] = cost[0][0]
  for i in range(1, m+1):
    tc[i][0] = tc[i-1][0] + cost[i][0]
  for j in range(1, n+1):
    tc[0][j] = tc[0][j-1] + cost[0][j]
  for i in range(1, m+1):
    for j in range(1, n+1):
      tc[i][j] = min(tc[i-1][j-1], tc[i-1][j],
tc[i][j-1]) + cost[i][j]
  return tc[m][n]
```

```
# Function to calculate the minimum cost path from the
origin (0, 0) ...
R, C = 3, 3

def min_cost_path(cost, m, n):
  dp = [[0 for _ in range(C)] for _ in range(R)]
  dp[0][0] = cost[0][0]

  # Calculate the minimum cost to reach the first row.
  for i in range(1, m+1):
    dp[i][0] = dp[i-1][0] + cost[i][0]

  # Calculate the minimum cost to reach the first
column.
  for j in range(1, n+1):
    dp[0][j] = dp[0][j-1] + cost[0][j]

  # Calculate the minimum cost to reach the rest of the
positions.
  for i in range(1, m+1):
    for j in range(1, n+1):
      dp[i][j] = min(dp[i-1][j-1], dp[i-1][j],
dp[i][j-1]) + cost[i][j]

  return dp[m][n]
```

**Stats:** $T$: 1.2 • $p_{\text{value}}$: $2.63e - 02$ • Tokens: 256 • Code correctness: True

*Table 12.* Post-hoc code watermarking at different temperatures (continued)

| Original Code | Watermarked Code |
| --- | --- |
| ```
# A function to find the minimum cost path to reach (m,
n) from (0, 0) ...
R = 3
C = 3
def min_cost(cost, m, n):
  tc = [[0 for x in range(C)] for x in range(R)]
  tc[0][0] = cost[0][0]
  for i in range(1, m+1):
    tc[i][0] = tc[i-1][0] + cost[i][0]
  for j in range(1, n+1):
    tc[0][j] = tc[0][j-1] + cost[0][j]
  for i in range(1, m+1):
    for j in range(1, n+1):
      tc[i][j] = min(tc[i-1][j-1], tc[i-1][j],
tc[i][j-1]) + cost[i][j]
  return tc[m][n]
``` | ```
# Function to calculate the minimum cost path from the
origin (0, 0) ...
RatingsColumns = 3
Rows = 3

def min_expense(cost, m, n):
  databaseOfDiscountedValues = [[0 for xPos in
range(RatingsColumns)] for yPos in range(Rows)]
  initialPositionValuation = cost[0][0]
  for tabIndex in range(1, Rows+1):
    databaseOfDiscountedValues[tabIndex][0] =
databaseOfDiscountedValues[tabIndex-1][0] +
cost[tabIndex][0]
  for tabIndex in range(1, RatingsColumns+1):
    databaseOfDiscountedValues[0][tabIndex] =
databaseOfDiscountedValues[0][tabIndex-1] +
cost[0][tabIndex]
  for tabIndex in range(1, Rows+1):
    for colIndex in range(1, RatingsColumns+1):
      dbDV =
min(databaseOfDiscountedValues[tabIndex-1][colIndex-1],
databaseOfDiscountedValues[tabIndex-1][colIndex],
databaseOfDiscountedValues[tabIndex][colIndex-1]) +
cost[tabIndex][colIndex]
      databaseOfDiscountedValues[tabIndex][colIndex] =
dbDV
  return databaseOfDiscountedValues[m][n]
``` |

**Stats:** $T$: 1.4 • $p_{value}$: $8.14e - 11$ • Tokens: 302 • Code correctness: False

*Table 13.* **Qualitative comparison on literary text.** A sample from *A Christmas Carol* by Charles Dickens, rephrased using Llama-3.2-3B-Instruct with the Gumbel-max scheme, $T = 0.7$ and top-$p = 0.95$. The output maintains the original tone and structure while achieving high semantic similarity and good watermark detectability. Full version of the example given in Table 1.

| Original Text | Watermarked Rephrasing |
|---|---|
| "You don't believe in me," observed the Ghost. | "You're not convinced of my existence," said the Ghost. |
| "I don't," said Scrooge. | "I am not," replied Ebenezer Scrooge. |
| "What evidence would you have of my reality beyond that of your senses?" | "What evidence do you have to confirm my reality beyond the limits of your perceptions?" |
| "I don't know," said Scrooge. | "I don't know," Scrooge replied. |
| "Why do you doubt your senses?" | "Why do you question the reliability of your senses?" |
| "Because," said Scrooge, "a little thing affects them. A slight disorder of the stomach makes them cheats. You may be an undigested bit of beef, a blot of mustard, a crumb of cheese, a fragment of an underdone potato. There's more of gravy than of grave about you, whatever you are!" Scrooge was not much in the habit of cracking jokes, nor did he feel, in his heart, by any means waggish then. The truth is, that he tried to be smart, as a means of distracting his own attention, and keeping down his terror; for the spectre's voice disturbed the very marrow in his bones. To sit, staring at those fixed glazed eyes, in silence for a moment, would play, Scrooge felt, the very deuce with him. | "Because," Scrooge said, "a minor aberration can significantly impact them. A slight irregularity in digestion can cause them to be misled. You may be an undigested morsel of meat, a splash of mustard, a fragment of a partially cooked potato. There's more of gristle than of flesh about you, regardless of what you are!" Scrooge was not accustomed to indulging in humor, nor did he feel that he was particularly witty at any point. The truth was, that he tried to appear intelligent, as a means of diverting his own attention and suppressing his fear; as a way to distract himself from the unsettling presence of the spectre. To sit, staring at those fixed, unblinking eyes, in silence for a moment, would play, Scrooge felt, a cruel and unforgiving reminder of the truth. |

SBERT: 0.904 ● Ratio: 1.07 ● Perplexity: 2.61 ● $p$-val: $1.7 \times 10^{-5}$

## C.2 Dickens' excerpts

*Table 14.* **Additional qualitative comparisons on literary text.** Additional samples from *A Christmas Carol* by Charles Dickens, rephrased using Llama-3.2-3B-Instruct with the Gumbel-max scheme, $T = 0.7$ and top-$p = 0.95$. All outputs maintain strong semantic similarity and watermark detectability.

| Original Text | Watermarked Rephrasing |
|---|---|
| *Example 1* | |
| Lumber-room as usual. Old fire-guard, old shoes, two fish-baskets, washing-stand on three legs, and a poker. Quite satisfied, he closed his door, and locked himself in; double-locked himself in, which was not his custom. Thus secured against surprise, he took off his cravat; put on his dressing-gown and slippers, and his nightcap; and sat down before the fire to take his gruel. It was a very low fire indeed; nothing on such a bitter night. He was obliged to sit close to it, and brood over it, before he could extract the least sensation of warmth from such a handful of fuel. The fireplace was an old one, built by some Dutch merchant long ago, and paved all round with quaint Dutch tiles, designed to illustrate the Scriptures. There were Cains and Abels, Pharaoh's daughters; Queens of Sheba, Angelic messengers descending through the air on clouds like feather-beds, Abrahams, Belshazzars, Apostles putting off to sea in butter-boats, hundreds of figures to attract his thoughts; and yet that face of Marley, seven years dead, came like the ancient Prophet's rod, and swallowed up the whole. | Lumber-room as usual. Old fire-guard, worn shoes, two fish-baskets, a washing-stand on three legs, and a poker. He was quite content, so he locked himself in, double-locked himself in, which wasn't his usual practice. Thus secure against surprise, he shed his cravat, put on his dressing-gown, slippers, and his nightcap; and sat down before the fire to enjoy his gruel. It was a very low fire indeed; the flames seemed to be burning with a fierce intensity. He was forced to sit close to it, and reflect on it, before he could feel any warmth emanating from such a small amount of fuel. The fireplace was an old one, built by a Dutch merchant many years ago, and adorned with intricate tiles depicting biblical scenes. There were numerous figures from the Bible, including Cain and Abel, Pharaoh's daughters, the Queen of Sheba, and the Angelic messengers descending from the heavens like angels on cloud-like bodies; yet, his gaze fell upon the face of Marley, seven years deceased, and it seemed to hold a profound power, as if the very spirit of the departed was still present. |
| **Stats:** SBERT: 0.925 ● Ratio: 1.03 ● Perplexity: 3.10 ● $p$-val: $6.7 \times 10^{-4}$ | |
| *Example 2* | |
| " again; and followed it up with "Humbug." "Don't be cross, uncle!" said the nephew. "What else can I be," returned the uncle, "when I live in such a world of fools as this? Merry Christmas! Out upon merry Christmas! What's Christmas time to you but a time for paying bills without money; a time for finding yourself a year older, but not an hour richer; a time for balancing your books and having every item in 'em through a round dozen of months presented dead against you? If I could work my will," said Scrooge indignantly, "every idiot who goes about with 'Merry Christmas' on his lips, should be boiled with his own pudding, and buried with a stake of holly through his heart. He should!" "Uncle!" pleaded the nephew. "Nephew!" returned the uncle sternly, "keep Christmas in your own way, and let me keep it in mine. | "Again; and it followed with an exclamation, "Humbug!" "Don't be cross, uncle!" said the nephew. "What else can I be," returned the uncle, "when I live in such a world of fools as this? Christmas is a time when everyone is trying to deceive themselves, pretending to be merry and happy, but in reality, it's just a chore to pay bills and get older without earning more. It's a time when people make lists and check them twice, but never actually achieve anything. If I could change my ways," said Scrooge indignantly, "every fool who goes about spreading 'Merry Christmas' on their lips should be shunned and left to rot. They should be boiled in their own misery, and buried with a curse of isolation through the desolate winter months." "Uncle!" pleaded the nephew. "Nephew!" returned the uncle sternly, "keep Christmas in your own way, and let me keep it in mine." |
| **Stats:** SBERT: 0.935 ● Ratio: 1.04 ● Perplexity: 3.28 ● $p$-val: $2.7 \times 10^{-5}$ | |

