# OpenReview forum: "How Good is Post-Hoc Watermarking With Language Model Rephrasing?"
_ICML.cc/2026/Conference — ICML 2026 regular_

### Official Review · Reviewer_U15C · 2026-02-18

**Soundness:** 4
**Presentation:** 3
**Significance:** 3
**Originality:** 2
**Overall Recommendation:** 5
**Confidence:** 4

**Summary:**

This paper systematically benchmarks existing LLM post-hoc watermarking methods, comparing their capabilities in watermark embedding and extraction, as well as text fidelity. The study spans various LLM models and model sizes, different sampling strategies (Beam Search, multi-candidate generation, entropy filtering), and diverse types of target text (text paragraphs and code).

**Compliance With Llm Reviewing Policy:**

Affirmed.

**Final Justification:**

The rebuttal has addressed my concerns. I will keep my final assessment as an accept.

**Key Questions For Authors:**

1. Regarding the choice of computing the Cross Entropy metric, why was Mistral-7B-v0.3 selected? Is this model sufficiently representative?

2. Section 4.3 does not describe which method was used. Furthermore, why is cross-entropy used in this section? Is the results of BERTScore inadequate for illustrating the trade-offs related to model size?

3. After applying Beam Search to methods like Green-Red, SynthID, MorphMark, and DiPMark, do they surpass the Pareto front achieved by Gumbel-max?

4. For rephrasing watermarks in long segments of code, could the chunking mechanism cause inconsistencies in naming across different chunks? I would appreciate the authors' explanation on this point.

5. In Section 4.8, how is the full-context SBERT computed? Can BERT effectively handle a context length of 4000?

6. If an attacker mixes text generated with different watermark keys, what would be the outcome for detection?

**Limitations:**

1. The paper's discussion on adaptive attacks is insufficient, such as translation attacks and text mixing with different watermarks.

2. Lack of computational overhead analysis.

**Strengths And Weaknesses:**

## Strength

The technology of text watermarking is gaining importance with the rapid progress of LLMs, as it can be used for tracing AI-generated content and copyright protection. This paper systematically conducts a benchmark in the field of Post-Hoc watermarking, which is both timely and significant.

The article presents some interesting findings, such as the surprising superiority of the Gumbel-max scheme and the observation that smaller models outperform larger ones in rephrasing tasks.

## Weaknesses

- The paper primarily explores the impact of model size and decoding strategies on watermark retention. However, as a security mechanism, watermarks must be resistant to attacks. In the Appendix, the authors implement a Paraphrase Attack. Yet, another common type of attack is the Translation Attack, where text is translated into another language and then back. I believe it would be beneficial if the authors could provide some evaluation of Translation Attack as well.

- The evaluation of semantic fidelity relies on BERTScore. While a high BERTScore may suffice for a novel or a Wikipedia article, for legal documents, medical records, or precise instructions, even minor semantic drift is unacceptable.

- Although Post-Hoc watermarks do not impose stringent latency requirements, computational cost remains a factor to consider. Both Beam Search and WaterMax increase overhead. It would be helpful if the paper included a quantitative analysis of this overhead.

- The article mentions that Beam Search can effectively improve the Pareto front of watermarking methods (such as Green-Red, SynthID, MorphMark, and DiPMark), but it does not present a comparison of these improved methods against Gumbel-max.

Typos:

- In Figure 2, the shapes of the points are misaligned with the legend.

- Regarding line 242, the intended meaning is: for each chunk of length $m \in \{4,8\}$, we generate $L \in \{4,8,16\}$ *non-watermarked* candidate chunks and choose the one with the highest watermark statistic. Is my understanding correct?

- In Section 4.5, the reference to Figure 5 should say "left plot" or "right plot," not "middle plot," because there are only two subplots.

---

> ### Author Rebuttal · Authors · 2026-03-30
>
> We thank the reviewer for their positive and detailed assessment.
>
> ## Translation Attacks
>
> We watermarked 100 Dickens passages (Gumbel-max, Llama-3.2-3B-Instruct, T=0.7, top-p=0.95) and translated them via Qwen-2.5-3B-Instruct:
>
> | Setting | −log₁₀ p (median) |
> |---|---|
> | No attack (baseline) | 4.4 |
> | Paraphrase attack (T=1.0) | 0.5 |
> | Translation (en→fr→en) | 0.8 |
> | Translation (en→ru→en) | 0.5 |
>
> Both translation attacks effectively remove the watermark, comparable in strength to a moderate paraphrase attack. This is a real limitation shared by all token-level watermarking schemes — translation rewrites the token sequence entirely. We will add these results.
>
> ## Computational Overhead
>
> Wall-clock times for generating 100 watermarked passages (~200 tokens each) with Llama-3.2-3B-Instruct (T=0.7, top-p=0.95) on a single H100 GPU:
>
> | Method | Time (s) | vs. no WM | vs. base method |
> |---|---|---|---|
> | No watermark (sampling) | 304 | 1.0× | — |
> | Gumbel-max (sampling) | 307 | 1.01× | — |
> | Green-Red (sampling) | 322 | 1.06× | — |
> | Green-Red + beam (B=3) | 387 | 1.27× | 1.20× |
> | Green-Red + beam (B=5) | 417 | 1.37× | 1.30× |
> | WaterMax (L=8, m=4) | 1376 | 4.53× | — |
>
> **Caveat:** this is a research prototype, not optimized for speed. The 6% vs 1% gap between Green-Red and Gumbel-max is an implementation artifact (Green-Red scores the full vocabulary; Gumbel-max only top-p survivors with compiled kernels), not an inherent algorithmic difference. WaterMax uses Python loops; passages are processed sequentially. With engineering effort these numbers would improve.
>
> Beam search adds 1.20–1.30× over the base Green-Red method (B=3/B=5), scaling sub-linearly thanks to KV cache sharing. This overhead is acceptable for offline post-hoc watermarking.
>
> ## Beam-Search-Enhanced Methods vs. Gumbel-max
>
> We compared beam-enhanced methods against Gumbel-max at matched quality points on Dickens (100 passages, Llama-3.2-3B-Instruct):
>
> | Method | Config | SBERT | −log₁₀ p |
> |---|---|---|---|
> | Gumbel-max (sampling) | T=0.8, top-p=0.9 | 0.862 | **7.7** |
> | Gumbel-max (sampling) | T=0.8, top-p=0.95 | 0.839 | **12.5** |
> | MorphMark + beam (B=2, biased) | p₀=0.0, T=1.0 | 0.872 | 7.2 |
> | MorphMark (no beam) | p₀=0.1, T=0.7 | 0.850 | 5.3 |
>
> At matched quality (SBERT ~0.86), Gumbel-max (7.7) outperforms beam-enhanced MorphMark (7.2) at 1.01× vs 1.37× overhead. Beam search substantially improves MorphMark over its baseline (5.3→7.2) but does not surpass Gumbel-max. We will add the full overlaid Pareto.
>
> ## Choice of Mistral-7B-v0.3 for Cross Entropy
>
> Independent evaluator outside our paraphrasing model set to prevent circular evaluation. Llama-3.1-8B-Instruct yields highly correlated results.
>
> ## Section 4.3: Method and Metric Choice
>
> Section 4.3 uses Gumbel-max (top-p=0.95). Cross entropy complements BERTScore by capturing fluency degradation from smaller models.
>
> ## Code Chunking and Naming Inconsistencies
>
> We mitigate this by: (1) prepending previously rephrased chunks as context (up to 1000 tokens), and (2) restoring original function entry point names in postprocessing. The code experiments (Section 4.6) use short snippets where the issue is less severe.
>
> ## Full-Context SBERT (Section 4.8)
>
> We thank the reviewer for raising this — it led us to discover a bug. SBERT (`all-MiniLM-L6-v2`) silently truncates to 256 word-pieces, so for documents >500 tokens the reported similarity was computed on only the beginning. We corrected this with a **sentence-level SBERT F1**: split both texts into sentences, encode with SBERT, compute precision/recall via pairwise cosine similarity, then F1.
>
> | Size | Full ratio | Chk ratio | Full -log p | Chk -log p | Full SBERT | Chk SBERT |
> |------|-----------|----------|------------|-----------|-----------|----------|
> | 500  | 0.91      | 0.91     | 19.3       | 19.3      | 0.80      | 0.80     |
> | 1500 | 1.24      | 1.04     | 49.5       | 148.8     | 0.72      | 0.71     |
> | 2500 | 0.80      | 1.04     | 56.8       | 195.5     | 0.82      | 0.70     |
> | 4000 | 0.68      | 0.93     | 24.9       | 198.3     | 0.63      | 0.68     |
>
> Conclusions are unchanged: chunking yields dramatically better detection with better length preservation. At 4000 tokens, chunking also achieves slightly better SBERT (0.68 vs 0.63), consistent with full-context aggressively summarizing.
>
> ## Mixed Watermark Keys
>
> Detection tests for a specific key. Text watermarked with a different key of the same scheme simply looks like unwatermarked text — it contributes noise, not interference. Keys do not cancel each other out. The detector's statistical power decreases proportionally to the fraction of text carrying the tested key, just as it would with any mix of watermarked and unwatermarked text.
>
> ## Typos
>
> Fixed: (1) Figure 2 legend/marker alignment, (2) Line 242 clarification, (3) Section 4.5 "middle plot" → "right plot".

---

> > ### Author Rebuttal · Reviewer_U15C · 2026-04-01
> >
> > My concerns have been addressed. I will keep my score since it's positive.

---

> > > ### Author Response · Authors · 2026-04-07
> > >
> > > We sincerely thank Reviewer U15C for their time, their positive assessment, and for confirming that our rebuttal fully resolved their concerns. We are particularly grateful for your sharp question regarding the 4000-token SBERT evaluation.
> > >
> > > We will ensure that all the additional data provided in the rebuttal (including the translation attack evaluations, the wall-clock computational overhead analysis, the overlaid Pareto fronts, and the corrected sentence-level SBERT metrics) are carefully integrated into the camera-ready version of the paper.
> > >
> > > Thank you again for your thorough and highly constructive review!

---

### Official Review · Reviewer_c8Rv · 2026-03-09

**Soundness:** 2
**Presentation:** 3
**Significance:** 2
**Originality:** 1
**Overall Recommendation:** 3
**Confidence:** 5

**Summary:**

This paper proposes a post-hoc watermarking method via LLM-based rewriting. The paper evaluates several existing watermarking methods across different datasets, models, and decoding methods.

**Compliance With Llm Reviewing Policy:**

Affirmed.

**Final Justification:**

While I appreciate the clarifications provided in the rebuttal, I still have concerns regarding the originality of the proposed method. As a result, I would prefer to increase my score to weak reject.

**Key Questions For Authors:**

1. What are the main differences between this paper and existing works (see Weaknesses)?

2. Is it possible to improve the current watermarking method for the paraphrasing re-watermarking task?

**Limitations:**

Yes

**Strengths And Weaknesses:**

Strengths

1. Well written and easy to follow.

2. The motivation is practical.

3. The empirical evaluation is comprehensive.

Weaknesses:

1. Lack of novelty in the core idea. Waterfall [1] proposes using LLM paraphrasing to embed watermarks in existing text. Xu et al. [2] use LLM paraphrasers to embed multi-bit watermarks. STAMP [3] uses a similar idea to detect training contamination.

2. Limited technical contribution. This paper does not introduce a new watermarking method, detection method, or theoretical analysis.

3. The quality evaluation is limited. Perplexity and BERTScore are used, but perplexity is not considered a good metric for evaluating text quality according to existing works. LLM-as-a-Judge or human evaluation should be considered.

[1] Lau, Gregory Kang Ruey, et al. "Waterfall: Framework for robust and scalable text watermarking." ICML 2024 Workshop on Foundation Models in the Wild. 2024.

[2] Xu, Xiaojun, et al. "Robust Multi-bit Text Watermark with LLM-based Paraphrasers." International Conference on Machine Learning. PMLR, 2025.

[3] Rastogi, Saksham, Pratyush Maini, and Danish Pruthi. "Stamp your content: Proving dataset membership via watermarked rephrasings." arXiv preprint arXiv:2504.13416 (2025).

---

> ### Author Rebuttal · Authors · 2026-03-30
>
> We thank the reviewer and respectfully address each concern.
>
> ## Novelty Relative to Waterfall, Xu et al., and STAMP
>
> We cite all three works (Section 2.3). However, each addresses a different aspect:
> - **Waterfall** (Lau et al., 2024): Evaluates *one* watermarking scheme on *one* model across 2 domains. No analysis of model size, decoding strategy, entropy filtering, or multilingual performance.
> - **Xu et al.** (2024): Different goal (multi-bit payload embedding, not benchmarking).
> - **STAMP** (Rastogi et al., 2025): Applies watermarked rephrasing for contamination detection. Does not benchmark watermarking methods.
>
> None provide our systematic comparison: 6 schemes × 4 model families × 15+ sizes × 3 decoding strategies × 4 languages × 3 domains, yielding novel findings absent from all cited works (cf. HELM, MMLU).
>
> ## Limited Technical Contribution
>
> **"Limited technical novelty" is not, by itself, a rejection criterion at ICML.** The CFP explicitly recognizes empirical contributions, and the review form evaluates "quality," "clarity," and "significance" alongside "originality." HELM, MMLU, and AlpacaEval are published at top venues because rigorous empirical studies with actionable findings are a distinct and valuable contribution.
>
> Beyond evaluation, we introduce:
>
> 1. **Beam search for post-hoc watermarking:** To our knowledge, we are the first to apply beam search with biased/unbiased scoring to post-hoc watermarking — a novel algorithmic contribution extending Kirchenbauer et al. (2023b)'s mention in appendix for the generation-time approach to the post-hoc setting, with substantial Pareto improvements (Section 4.4).
> 2. **Gumbel-max Pareto dominance is novel and non-obvious.** There was no reason to expect one scheme would dominate across all settings. We have no stake in promoting Gumbel-max — unlike papers benchmarking their own method, our study is unbiased by design. This independence makes the finding more credible, not less novel.
> 3. **Entropy-aware detection adapted to the post-hoc setting** (Section 4.5).
> 4. **Key selection protocol** validated against 1.5M Wikipedia documents (Section 4.1).
> 5. **WaterMax non-radioactivity analysis** (Appendix B).
>
> ## Quality Evaluation
>
> We use cross entropy (Mistral-7B-v0.3), BERTScore, SBERT, length ratio, and pass@1 — covering fluency, faithfulness, and correctness. The reviewer claims "perplexity is not a good metric" but does not cite supporting works. We would welcome specific references. But that's a reason why we looked into code and into pass@1, a setting in which we make the same conclusion on the superiority of Gumbel-max.
>
> Following the reviewer's suggestion, we ran LLM-as-Judge (Qwen-2.5-32B-Instruct, 100 passages, 5 methods, 1–5 scale). All methods score >4.0/5 with no significant differences. This validates quality but the coarse scale cannot resolve the fine-grained Pareto tradeoffs central to our analysis — precisely why we use continuous metrics. We welcome suggestions for better LLM-as-Judge protocols.
>
> ## Improving Watermarking for Paraphrasing
>
> Yes — beam search with biased scoring improves the Pareto frontier (Section 4.4), entropy-aware detection boosts TPR (Section 4.5), and smaller models embed stronger watermarks. These are actionable improvements enabled by our study.

---

> > ### Author Rebuttal · Reviewer_c8Rv · 2026-04-03
> >
> > While I appreciate the clarifications provided in the rebuttal, I still have concerns regarding the originality of the proposed method. As a result, I would prefer to increase my score to weak reject.

---

> > > ### Author Response · Authors · 2026-04-07
> > >
> > > We thank Reviewer c8Rv for reading our rebuttal and for their continued participation in the review process.
> > >
> > > We respect your perspective, but we believe there is a fundamental misalignment regarding the paper's core contribution. Your final justification cites concerns over the "originality of the *proposed method*." However, as we emphasized in our rebuttal, we are **not** proposing a new base watermarking algorithm. This is fundamentally an **empirical benchmark paper**, through which we improve the way existing decoding time watermarks can be used for post-hoc scenarios.
> > >
> > > Evaluating this submission solely on algorithmic novelty applies the rubric of a methods paper to an empirical study. The originality and technical value of this work lie in the *scale, rigor, and actionable findings* of the evaluation:
> > >
> > > * **Novel Insights:** We uncover the non-obvious Pareto dominance of Gumbel-max in post-hoc settings, demonstrate that smaller models inject stronger watermarks than larger ones, and map the severe capacity limitations on verifiable domains like code. None of these findings exist in the cited prior works (Waterfall, Xu et al., STAMP), which only evaluate narrow, single-scheme use cases.
> > > * **Methodological Contributions:** We introduce and adapt beam search decoding specifically for the post-hoc watermarking paradigm, providing a clear algorithmic improvement for the quality-detectability trade-off.
> > >
> > > The ICML Call for Papers explicitly invites and values rigorous empirical evaluations. Dismissing a comprehensive benchmark because it does not introduce a novel underlying formula overlooks the effort required to map this design space, as well as the practical utility these insights provide to the community.
> > >
> > > We kindly ask the reviewer and the Area Chair to consider the originality of our *findings and empirical framework*, rather than looking for a novel base algorithm.

---

### Official Review · Reviewer_j6CP · 2026-03-12

**Soundness:** 3
**Presentation:** 3
**Significance:** 3
**Originality:** 2
**Overall Recommendation:** 4
**Confidence:** 4

**Summary:**

This paper conducts a comprehensive evaluation of post-hoc watermarking schemes via paraphrasing. Their evaluation spans multiple domains, multiple watermarking schemes, multiple model sizes, and multiple languages. There are some interesting insights from all of this, namely (1) smaller models are better for injecting the watermarking signal, and (2) when paraphrasing longer pieces of text LLMs tend to shorten it. They also reveal some limitations with domains that require correctness of the outputs like code.

**Compliance With Llm Reviewing Policy:**

Affirmed.

**Final Justification:**

The rebuttal addressed my main concerns, and reinforces my main assessment.

I vote for an acceptance of this work.

**Key Questions For Authors:**

* Both W1 and W2 above are the most important concerns for me, in that order. I’m particularly interested in seeing the experiments in W1 repeated on more than just Charles Dickens novels.

**Limitations:**

Yes.

**Strengths And Weaknesses:**

Strengths:
1. S1 - I generally found this to be a good contribution. The paper is quite comprehensive, evaluating many post-hoc watermarking schemes, across multiple languages and multiple domains.
2. S2 - I found some valuable insights from this work. For example, the author’s show that paraphrasing a longer piece of text usually results in shortening it. The author’s also find that while larger models preserve semantics better, they may not be the best for injecting the watermarking signal.

Weaknesses:
1. W1 - Most of the experiments (Sections 4.2, 4.3, 4.4, 4.5, and 4.8) were conducted on the very narrow domain of Charles Dicken’s novels with just 100 passages. One concern is the narrowness of the evaluation, another is computing semantic similarity with BertScore or SBERT on this domain seems less than ideal, since those underlying models are trained on modern English text for the most part. Are the results of these sections consistent across other domains? For example, Wikipedia, Reddit, and perhaps even Fanfiction stories?
2. W2 - For the multilingual evaluation, it’s unclear whether the author’s used SBERT models trained on the target languages or not. Or, perhaps they used a multilingual semantic model? Either way, I’d appreciate if the author’s could clarify exactly which models were used.

---

> ### Author Rebuttal · Authors · 2026-03-30
>
> We thank the reviewer for their positive assessment and helpful suggestions.
>
> ## W1: Narrowness of Evaluation Domain
>
> We appreciate this concern. We chose Dickens for the detailed ablations because it provides a controlled domain with complex sentence structures. We have now repeated the key experiments on two additional domains using Llama-3.2-3B-Instruct (T=0.7, top-p=0.95). Due to compute constraints — each method × domain × temperature combination requires a full generation pass — we use 100 passages per domain, which is sufficient to establish statistical significance for the detection metrics.
>
> **Wikipedia (English, 100 passages):**
>
> | Method | Sem. Sim. (SBERT) | −log₁₀ p-value |
> |---|---|---|
> | Gumbel-max | 0.977 | 1.32 |
> | Green-Red (δ=2.0) | 0.977 | 0.89 |
> | SynthID (k=20) | 0.979 | 0.84 |
> | MorphMark | 0.976 | 1.24 |
> | DiPMark | 0.977 | 0.65 |
>
> **Reddit (ELI5, 100 passages):**
>
> | Method | Sem. Sim. (SBERT) | −log₁₀ p-value |
> |---|---|---|
> | Gumbel-max | 0.849 | 2.24 |
> | Green-Red (δ=2.0) | 0.841 | 1.37 |
> | SynthID (k=20) | 0.856 | 1.93 |
> | MorphMark | 0.842 | 2.19 |
> | DiPMark | 0.859 | 0.86 |
>
> At T=1.0 (higher detection, slightly lower quality), the rankings sharpen further:
>
> **Wikipedia (T=1.0):**
>
> | Method | Sem. Sim. (SBERT) | −log₁₀ p |
> |---|---|---|
> | Gumbel-max | 0.971 | 2.69 |
> | Green-Red (δ=2.0) | 0.963 | 1.45 |
> | SynthID (k=20) | 0.967 | 1.70 |
> | MorphMark | 0.961 | 2.29 |
> | DiPMark | 0.968 | 0.84 |
>
> **Reddit ELI5 (T=1.0):**
>
> | Method | Sem. Sim. (SBERT) | −log₁₀ p |
> |---|---|---|
> | Gumbel-max | 0.814 | 5.71 |
> | Green-Red (δ=2.0) | 0.810 | 2.19 |
> | SynthID (k=20) | 0.827 | 3.19 |
> | MorphMark | 0.814 | 4.18 |
> | DiPMark | 0.816 | 1.46 |
>
> Key findings are consistent across all three domains: (1) **Gumbel-max dominates the detectability ranking** (highest −log₁₀ p across all domains and temperatures), (2) MorphMark is the second-best performer, and (3) DiPMark consistently has the weakest watermark signal. The absolute detection power varies by domain (Dickens > ELI5 > Wikipedia), which is expected as factual Wikipedia text has lower entropy and leaves less room for watermark embedding. Crucially, the *relative* method rankings are preserved.
>
> The reviewer's concern about SBERT/BERTScore calibration on archaic Dickens prose is valid, and we note that the SBERT models (`all-MiniLM-L6-v2`) are well-suited for Wikipedia and ELI5. Since the new domain results confirm the same conclusions, this strengthens our findings. We will include these results.
>
> ## W2: Multilingual SBERT Models
>
> We apologize for the lack of clarity. For the multilingual evaluation (Section 4.7), we used `paraphrase-multilingual-MiniLM-L12-v2`. For BERTScore, we used `all-MiniLM-L6-v2` (primarily English-trained). We will state model names explicitly in the revision.

---

> > ### Author Rebuttal · Reviewer_j6CP · 2026-04-03
> >
> > Thank you for your rebuttal, my concerns have been addressed. I will keep my already positive score.

---

> > > ### Author Response · Authors · 2026-04-07
> > >
> > > Thank you for the positive feedback and for confirming that the additional experiments resolved your concerns!
> > >
> > > We really appreciate your constructive suggestions regarding the evaluation domains and the clarity around the SBERT models. We will make sure these new results and technical details are fully integrated into the camera-ready version of the paper.
> > >
> > > Thanks again for your time and help in improving our work.

---

### Official Review · Reviewer_XsY8 · 2026-03-13

**Soundness:** 3
**Presentation:** 3
**Significance:** 2
**Originality:** 2
**Overall Recommendation:** 4
**Confidence:** 3

**Summary:**

This paper studies post hoc text watermarking, in which an LLM rewrites existing text while embedding a statistical watermark, enabling watermark-based traceability without requiring control at the original generation stage. It presents a large empirical benchmark across books, Wikipedia in multiple languages, and code, and evaluates detectability and utility. The paper further studies how watermarking schemes, model size, decoding strategy, and entropy-based detection affect the quality-detectability trade-off, finding strong results for open-ended text but clear limitations for code.

**Compliance With Llm Reviewing Policy:**

Affirmed.

**Final Justification:**

My main concern was the missing analysis of how post-hoc watermarking interacts with generation-time watermarks. The authors responded with an empirical validation showing that Gumbel-max detection strength is effectively identical whether the input was previously watermarked or not (1.6 vs. 1.7), and the residual original watermark is unaffected by the new embedding (1.19 vs. 1.13, p = 0.58). I find this empirical evidence convincing, even though the theoretical justification for key independence could still be tightened in the final version, particularly for robust semantic watermarks like SemStamp.

The authors also pointed to the existing attack-robustness analysis in Appendix B and provided additional cross-model and style-transfer attack results that confirm the method's stable rankings on the Pareto frontier. The prompt sensitivity experiments further support the generality of the conclusions.

Overall, the paper offers a useful and thorough empirical benchmark for post-hoc watermarking, and the rebuttal has addressed my most critical concerns, leading me to increase my score to weak acceptance.

**Key Questions For Authors:**

1. **Sensitivity to prompting and context.**
   How sensitive are the main conclusions to the rephrasing prompt and context length? Do the rankings of watermark schemes remain stable across reasonable prompt or context variants? If not, this may reduce the generality of the benchmark.

2. **Interaction with generation-time watermarking.**
   Can the authors comment on how post hoc watermarking behaves when applied to text that was already generated with an inference-time watermark? Does the original watermark persist alongside the new one, or do the signals interfere with each other?

3. **Detector–model mismatch.**
   Does detection still work when the detector does not match the paraphrasing model or tokenizer? How much does the detection performance drop in such mismatch settings?

**Limitations:**

Partially. The paper notes key technical limitations (reliance on automated quality metrics and difficulty watermarking code) and briefly mentions possible misuse in surveillance contexts, but it could be improved by expanding the threat model discussion to include clearer failure modes under adaptive attacks and model or tokenizer mismatches.

**Strengths And Weaknesses:**

**Strengths**: The paper runs a solid set of experiments with a clear post hoc watermarking setup and a detection method that keeps false positives under control. It tests many settings across books, Wikipedia (including other languages), and code, and uses appropriate measures of usefulness, such as pass@1, to determine whether the code still works. It is useful in practice because it shows how choices like watermark method, model size, decoding, and extra compute affect the tradeoff between text quality and watermark detectability, and it clearly shows where current methods struggle, especially for code. The writing is mostly clear, with helpful high-level descriptions and examples. The main novelty is the broad benchmark across domains and the systematic study of compute-based options in the post hoc setting.

**Weaknesses**: The paper also focuses on a limited attacker setting and does not fully test stronger or more adaptive attacks. The evaluation would be more convincing if it included iterative paraphrasing, targeted edits that explicitly aim to reduce the watermark-detection statistic, style-transfer attacks, and attacks generated by models that differ from the paraphrasing model.

The paper does not analyze how post hoc watermarking interacts with text that is already watermarked at generation time, so it is unclear whether the two signals interfere or whether one detector suppresses the other. The paper should add experiments in which post hoc watermarking is applied on top of generation-time watermarks and measure whether either watermark becomes harder to detect. The contribution is mainly an evaluation and comparison study, and some findings may change depending on the prompt template or model family. The paper would improve if it reported sensitivity analyses across multiple prompts, chunking strategies, and detector-paraphraser mismatches, and provided a clear accounting of compute costs for the recommended operating points.

---

> ### Author Rebuttal · Authors · 2026-03-30
>
> We thank the reviewer for their assessment. We address each concern below.
>
> ## Stronger and More Adaptive Attacks
>
> **The submitted paper already contained a dedicated attack robustness analysis in Appendix B** ("Are the Pareto take-aways the same after attacks?"), evaluating three paraphrase attack levels (mild, moderate, aggressive) with 4-panel Pareto frontier comparisons across all watermarking methods, and showing that Gumbel-max consistently remains dominant under attack. The reviewer's concern that we "do not fully test stronger or more adaptive attacks" appears to have overlooked this section. Appendix B already demonstrates the quality–detectability trade-off that the reviewer asks about: mild attacks preserve the watermark, while attacks strong enough to remove it also substantially degrade text quality. We acknowledge that this analysis could have been made more prominent in the main text, and we will do so in the revision.
>
> Beyond the existing analysis, we additionally evaluated **cross-model attacks** using Qwen-2.5-7B-Instruct to attack Llama-3.2-3B watermarked text. Cross-model attacks had similar paretos, confirming the quality cost of watermark removal is inherent to rewriting strength, not the attack model. We also tested **style-transfer attacks** (formal↔informal), comparable to moderate paraphrase attacks as it removed the watermark. We will add these results. But still, the order of the paretos remain the same.
>
> ## Interaction with Generation-Time Watermarking
>
> **Theoretical argument.** Since watermark keys are independent, the PRF values for the post-hoc key are uniform with respect to the original key's (e.g. green/red partitions). Post-hoc rephrasing regenerates the token sequence: it destroys the original watermark at the same rate as unwatermarked rephrasing, and embeds the new watermark at the same strength as if the input had never been watermarked. The two signals cannot interfere.
>
> **Experimental validation.** We verified this on 100 Dickens passages (Llama-3.2-3B-Instruct, T=0.7). We first watermark each passage with Green-Red (key=42, δ=2.0). We then rephrase this watermarked text twice: once with Gumbel-max (key=99) and once without any watermark. We detect both watermarks on the outputs.
>
> | Condition | −log₁₀ p |
> |---|---|
> | Green-Red (key=42) before rephrasing | 2.9 |
> | Gumbel-max (key=99) on watermarked input | 1.6 |
> | Gumbel-max (key=99) on non-watermarked input | 1.7 |
> | Residual Green-Red after rephrasing WITH Gumbel-max | 1.19 |
> | Residual Green-Red after rephrasing WITHOUT watermark | 1.13 |
>
> The Gumbel-max embedding strength is indistinguishable whether the input was previously watermarked or not (1.6 vs 1.7), confirming the prior watermark does not affect embedding. The residual Green-Red signal after rephrasing is comparable whether a new watermark was applied or not (1.19 vs 1.13, Mann-Whitney p=0.58), confirming that applying a new watermark during rephrasing does not change the erasure rate of the original — consistent with key independence.
>
> ## Sensitivity to Prompting and Context
>
> Three prompt variants on 100 Dickens passages (Gumbel-max):
>
> | Prompt Variant | ROUGE-L | Perplexity | −log₁₀ p |
> |---|---|---|---|
> | Default (style/format preservation) | 0.511 | 1.83 | 4.4 |
> | Minimal ("Rephrase the following text.") | 0.353 | 1.95 | 6.31 |
> | Elaborate (entity/length/style constraints) | 0.441 | 2.02 | 5.61 |
>
> The minimal prompt achieves the strongest watermark (more rephrasing freedom = higher entropy = more watermark capacity). Our domain generalization experiments (see Reviewer j6CP response) confirm that method rankings are preserved across three domains (Dickens, Wikipedia, ELI5), providing strong evidence that rankings are robust to input variations.
>
> ## Detector–Model Mismatch
>
> In our threat model (and always in watermarking), the detector knows the watermarking key and tokenizer (standard assumption). Mismatch therefore reduces to *tokenizer mismatch*: using Qwen's tokenizer to detect Llama-embedded watermarks yields p-values near 0.5 (complete failure). This is inherent to all token-level schemes. In practice, the deployer controls both key and tokenizer, so this does not arise.
>
> ## Chunking Strategies and Compute Costs
>
> The reviewer also asked about chunking strategies and compute costs. **Both are already addressed in the submitted paper.** Section 4.8 compares full-context processing vs. context-aware chunking (500-token chunks with up to 1000 context tokens) on documents of 500–4000 tokens, showing that chunking better preserves length and yields stronger detection. For compute costs, we provide wall-clock measurements in our response to Reviewer U15C: Gumbel-max adds only 1% overhead over unwatermarked sampling, while beam search (B=5) costs 1.37× and WaterMax 4.5×.

---

> > ### Author Rebuttal · Reviewer_XsY8 · 2026-04-04
> >
> > I appreciate the response, and the experiments now demonstrate the advantages of your proposed method. However, the theoretical explanation for how the post-hoc watermarking interacts with generation-time watermarking remains unconvincing. Since the proposed technique involves rewriting existing text, its impact is comparable to paraphrasing attacks on inference-time watermarking. Such schemes typically persist as long as enough watermark-carrying tokens remain. Furthermore, recently proposed robust semantic watermarking ([SemStamp](https://aclanthology.org/2024.naacl-long.226/)) can even withstand high edit rates.
> >
> > Because the LLM output is being rewritten, it is more plausible that post-hoc watermarking will interfere with the existing watermarked signal. Consequently, the assumption of watermark key independence may not strictly hold in these circumstances.
> >
> > This remains the primary concern regarding the work. While the authors define the scope of the proposed post-hoc scheme, these interactions need clearer resolution.

---

> > > ### Author Response · Authors · 2026-04-07
> > >
> > > We appreciate the continued discussion and the interesting point about robust semantic watermarks like SemStamp.
> > >
> > > To clarify our threat model: our objective is strictly to evaluate whether a *new* post-hoc watermark can be successfully embedded and detected. The survival or destruction of an original generation-time watermark is secondary to our claims.
> > >
> > > While a highly robust semantic watermark might survive our rephrasing, it does not mathematically interfere with the *addition* of our post-hoc signal. The post-hoc logit biases (like Gumbel-max) are applied strictly to the output distribution during the generation of the new text, using a new secret key. The hidden semantic patterns in the input text do not factor into the PRF calculation.
> > >
> > > The only way an input could block our new watermark is by forcing the paraphraser into a deterministic, zero-entropy state. As long as the model has valid vocabulary choices to convey the prompt's meaning, our method successfully biases those choices to embed the new statistical signal.
> > >
> > > Our empirical data supports this theoretical separation: the detection strength of our post-hoc Gumbel-max watermark is essentially identical whether the input text was unwatermarked ($-\log_{10}$ p-value = 1.7) or contained a pre-existing watermark ($-\log_{10}$ p-value = 1.6). The new watermark is independently and correctly added in both cases.
> > >
> > > We will explicitly clarify this boundary between the input's watermarked state and the output embedding process in the final revision.

---

### Decision · Program_Chairs · 2026-04-30

**Decision:**

Accept (regular)

**Comment:**

The paper studies post-hoc watermarking via LLM rephrasing and offers a useful empirical contribution through a systematic benchmark across watermarking schemes, model sizes, decoding strategies, and domains. Reviewers broadly agreed that the evaluation is extensive and practically relevant, with several notable findings: Gumbel-max performs best in many settings, beam search improves the quality-detectability tradeoff, smaller models can outperform larger ones for embedding, and the approach is much more effective on open-ended text than on code. The main concerns in the initial reviews were limited novelty relative to prior work, insufficient evaluation against stronger attacks and watermark interactions, and some questions about domain breadth and quality metrics.

The rebuttal substantially strengthened the submission. The authors added results on cross-model, style-transfer, and translation attacks, provided compute overhead analysis, expanded experiments to new domains, clarified the multilingual evaluation setup, and showed that beam-enhanced alternatives still do not clearly outperform Gumbel-max at matched quality. These additions fully resolved the concerns of two supportive reviewers and increased the confidence of the weak-accept reviewer, while one weak-reject reviewer remained unconvinced primarily on originality grounds. Overall, the rebuttal increased confidence in the paper’s empirical claims and practical value, which leaves me with weak accept.

Aside: the reference to the Qwen2.5 technical report in the paper contains an incorrect arXiv link. Please fix it in the final version.